# Overcoming Slow Decision Frequencies in Continuous Control: Model-Based Sequence Reinforcement Learning for Model-Free Control

**Devdhar Patel & Hava T. Siegelmann**
Manning College of Information and Computer Sciences
University of Massachusetts
Amherst, MA 01002 USA
{devdharpatel, hava}@umass.edu

## Abstract

Reinforcement learning (RL) is rapidly reaching and surpassing human-level control capabilities. However, state-of-the-art RL algorithms often require timesteps and reaction times significantly faster than human capabilities, which is impractical in real-world settings and typically necessitates specialized hardware. We introduce Sequence Reinforcement Learning (SRL), an RL algorithm designed to produce a sequence of actions for a given input state, enabling effective control at lower decision frequencies. SRL addresses the challenges of learning action sequences by employing both a model and an actor-critic architecture operating at different temporal scales. We propose a "temporal recall" mechanism, where the critic uses the model to estimate intermediate states between primitive actions, providing a learning signal for each individual action within the sequence. Once training is complete, the actor can generate action sequences independently of the model, achieving model-free control at a slower frequency. We evaluate SRL on a suite of continuous control tasks, demonstrating that it achieves performance comparable to state-of-the-art algorithms while significantly reducing actor sample complexity. To better assess performance across varying decision frequencies, we introduce the Frequency-Averaged Score (FAS) metric. Our results show that SRL significantly outperforms traditional RL algorithms in terms of FAS, making it particularly suitable for applications requiring variable decision frequencies. Furthermore, we compare SRL with model-based online planning, showing that SRL achieves comparable FAS while leveraging the same model during training that online planners use for planning.

## 1 Introduction

Biological and artificial agents must learn behaviors that maximize rewards to thrive in complex environments. Reinforcement learning (RL), a class of algorithms inspired by animal behavior, facilitates this learning process (Sutton & Barto, 2018). The connection between neuroscience and RL is profound. The Temporal Difference (TD) error, a key concept in RL, effectively models the firing patterns of dopamine neurons in the midbrain (Schultz et al., 1997; Schultz, 2015; Cohen et al., 2012). Additionally, a longstanding goal of RL algorithms is to match and surpass human performance in control tasks (OpenAI et al., 2019; Schrittwieser et al., 2020; Kaufmann et al., 2023b; Wurman et al., 2022a; Vinyals et al., 2019; Mnih et al., 2015).

However, most of these successes are achieved by leveraging large amounts of data in simulated environments and operating at speeds orders of magnitude faster than biological neurons. For example, the default timestep for the Humanoid task in the MuJoCo environment (Todorov et al., 2012) in OpenAI Gym (Towers et al., 2023) is 15 milliseconds. In contrast, human reaction times range from 150 milliseconds (Jain et al., 2015) to several seconds for complex tasks (Limpert, 2011). Table 1 shows the significant gap between AI and humans in terms of timestep and reaction times. When RL

agents are constrained to human-like decision frequencies, even state-of-the-art algorithms struggle to perform in simple environments (Dulac-Arnold et al. (2021), Figure 5 in Appendix).

| Environment / Task | Timestep / Reaction Time |
|---|---|
| Inverted Pendulum | 40ms |
| Walker 2d | 8ms |
| Hopper | 8ms |
| Ant | 50ms |
| Half Cheetah | 50ms |
| Dota 2 1v1 (OpenAI et al., 2019) | 67ms |
| Dota 2 5v5 (OpenAI et al., 2019) | 80ms |
| GT Sophy (Wurman et al., 2022b) | 23-30ms |
| Drone Racing (Kaufmann et al., 2023a) | 10ms |
| Humans | $\geq$ 150ms |

Table 1: Timestep / reaction times for various benchmark environments and popular works that pit humans vs. AI.

The primary reason for this difficulty is the implicit assumption in RL that the environment and the agent operate at a constant timestep. Consequently, in embodied agents that implement RL algorithms, all components: sensors, compute units, and actuators—are synchronized to the same frequency at the algorithmic level. Typically, this frequency is limited by the speed of computation in artificial agents (Katz et al., 2019). As a result, robots often require fast onboard computing hardware (CPU or GPU) to achieve higher control frequencies (Margolis et al., 2024; Li et al., 2022; Haarnoja et al., 2024).

To allow the RL agent to observe and react to changes in the environment quickly, RL algorithms are forced to set a high frequency. Even in completely predictable environments, when the agent learns to walk or move, a small timestep is required to account for the actuation frequency required for the task, but it is not necessary to observe the environment as often or compute new actions as frequently. RL algorithms suffer from catastrophic failure due to missing inputs (also referred to as observational dropout). This behavior level gap between RL and humans can be bridged by bridging the gap in the underlying process.

Towards that end, we propose Sequence Reinforcement Learning (SRL), a model for action sequence learning based on the role of the basal ganglia (BG) and the prefrontal cortex (PFC). Our model learns open-loop control utilizing a low decision frequency. Additionally, the algorithm utilizes a simultaneously learned model of the environment during its training but can act without it for fast and cheap inference. We demonstrate the algorithm achieves competitive performance on difficult continuous control tasks while utilizing a fraction of observations and calls to the policy. To the best of our knowledge, SRL is the first to achieve this feat. To further quantify this result and set a benchmark for control at slow frequencies, we introduce the Frequency Averaged Score (FAS) and demonstrate that SRL achieves significantly higher FAS than Soft-Actor-Critic (SAC) (Haarnoja et al., 2019) and Generative-Planning-Method (GPM) (Zhang et al., 2022). Additionally, we demonstrate that on complex environments (with high state and action dimensions), SRL also beats model-based online planning on FAS. Finally, in the appendix, we discuss the available evidence in neuroscience that has inspired our algorithm and also present promising initial results in the proposed future work of generative replay in latent space.

## 2 NECESSITY OF SEQUENCE LEARNING: FREQUENCY, DELAY AND RESPONSE TIME

To perform any control task, the agent requires the following three components: Sensor, Processor/Computer, Actuator. In the traditional RL framework, all three components act at the same frequency due to the common timestep. However, this is not the case in biological agents that have different sensors of varying frequencies that are often faster than the compute frequency or the speed at which the brain can process the information (Borghuis et al., 2019). Additionally, in order to afford fast and precise control, the actuator frequency is also much faster than the compute frequency (see Figure 9 in Appendix).

Low-compute hardware faces two primary challenges for real-time control: delay and throughput. The high inference times associated with low-compute devices result in a delay between receiving observations and performing corresponding actions in the environment. Additionally, they lead to low decision frequencies in sequential decision-making tasks.

While many prior works have focused on addressing delay by designing delay-aware algorithms (Chen et al., 2020; 2021; Derman et al., 2021), mitigating delay alone does not resolve the performance issues caused by low decision frequency. Adapting RL algorithms to operate effectively in low-frequency compute settings remains an open challenge (Dulac-Arnold et al., 2021).

The Sequence Reinforcement Learning (SRL) algorithm offers a promising solution to these low-decision frequency scenarios. To address the complete set of challenges posed by low-compute environments, SRL can be integrated with delay-aware algorithms to simultaneously manage delays while achieving higher action frequencies. Moreover, SRL inherently addresses delays by producing sequences of actions that can bridge the gap caused by processing latency. For example, if output arrives with a delay of $n$ timesteps, the first $n$ actions of the new sequence can be ignored, as they were already executed as part of the previous sequence. This mechanism ensures smooth and continuous action execution despite processing delays.

**Why low-frequency compute?**

Recent advancements in reinforcement learning (RL) algorithms, combined with high-speed computing, have led to two common approaches for addressing the speed-accuracy trade-off:

1. **Faster hardware:** The use of GPUs has become standard for enabling rapid inference in autonomous agents (Long et al., 2024; Csomay-Shanklin et al., 2024; Lazcano, 2024). However, GPUs are often impractical in many real-world applications due to their high cost, energy demands, and large physical size. As a result, recent research has also focused on developing specialized embedded deep learning accelerators to address these challenges (Akkad et al., 2023).

2. **Software optimization:** Techniques such as quantization (Jafarpourmarzouni et al., 2024), multi-exit networks (Rahmath P et al., 2022), and model compression (Neill, 2020) are commonly employed to reduce inference times without requiring additional hardware.

In essence, these approaches focus on either accelerating hardware or optimizing software. In this work, we propose an alternative paradigm: enhancing accuracy at low operating frequencies instead of striving for high frequencies. By advancing research in this direction, we aim to relax the dependency on high-performance hardware, enabling RL algorithms to operate effectively on low-compute devices while also making ultra-high-frequency control feasible on current hardware platforms.

## 3 Related Work

### 3.1 Model-Based Reinforcement Learning

Model-Based Reinforcement Learning (MBRL) algorithms leverage a model of the environment, which can be either learned or known, to enhance RL performance (Moerland et al., 2023). Broadly, MBRL algorithms have been utilized to:

1. Improve Data Efficiency: By augmenting real-world data with model-generated data, MBRL can significantly enhance data efficiency (Yarats et al., 2021; Janner et al., 2019; Wang et al., 2021).

2. Enhance Exploration: MBRL aids in exploration by using models to identify potential or unexplored states (Pathak et al., 2017; Stadie et al., 2015; Savinov et al., 2019).

3. Boost Performance: Better learned representations from MBRL can lead to improved asymptotic performance (Silver et al., 2017; Levine & Koltun, 2013).

4. Transfer Learning: MBRL supports transfer learning, enabling knowledge transfer across different tasks or environments (Zhang et al., 2018; Sasso et al., 2023).

5. Online Planning: Models can be used for online planning with a single-step policy (Fickinger et al., 2021). However, this approach increases model complexity, as each online planning step necessitates an additional call to the model. This makes it unsuitable for applications with limited computational budgets and strict requirements for fast inference.

Compared to online planning, our algorithm maintains a model complexity of zero after training, eliminating the need for any model calls post-training for generating a sequence of actions. This significantly reduces the computational and energy requirements, making it more suitable for practical applications in constrained environments. Additionally, model-based online planning is less biologically plausible than SRL. Wiestler & Diedrichsen (2013) demonstrated that the activations in the motor cortex reduce after skill learning, suggesting that the brain gets more efficient at performing the task after learning. In contrast, model-based online planning does not reduce in the compute and model complexity, but rather might increase in complexity as we perform longer sequences. SRL, on the other hand, has a model complexity of zero after training and thus is biologically plausible based on this observed phenomenon.

## 3.2 Model Predictive Control

Similar to model-based reinforcement learning, Model Predictive Control (MPC) utilizes a model of the system to predict and optimize future behavior. In the context of modern robotics, MPC has been effectively applied to trajectory planning and real-time control for both ground and aerial vehicles. MPC has been applied to problems like autonomous driving (Gray et al., 2013) and bipedal control (Galliker et al., 2022). Similar to online planning, MPC often requires access to a model of the system after training.

Additionally, similar to current RL, MPC requires very fast operational timesteps for practical applications. For example, Galliker et al. (2022) implemented a walker at 10 ms, Farshidian et al. (2017) implemented a four-legged robot at 4 ms, and Di Carlo et al. (2018) implemented the MIT Cheetah 3 at 33.33 ms.

## 3.3 Macro-Actions, Action Repetition, and Frame-skipping

Reinforcement Learning (RL) algorithms that utilize macro-actions demonstrate many benefits, including improved exploration and faster learning (McGovern et al., 1997). However, identifying effective macro-actions is a challenging problem due to the curse of dimensionality, which arises from large action spaces. To address this issue, some approaches have employed genetic algorithms (Chang et al., 2022) or relied on expert demonstrations to extract macro-actions (Kim et al., 2020). However, these methods are not scalable and lack biological plausibility. In contrast, our approach learns macro-actions using the principles of RL, thus requiring little overhead while combining the flexibility of primitive actions with the efficiency of macro-actions.

To overcome the curse of dimensionality while gaining the benefits of macro-actions, many approaches utilize frame-skipping and action repetition, where macro-actions are restricted to a single primitive action that is repeated. Frame-skipping and action repetition serve as a form of partial open-loop control, where the agent selects a sequence of actions to be executed without considering the intermediate states. Consequently, the number of actions is linear in the number of time steps (Kalyanakrishnan et al., 2021; Srinivas et al., 2017; Biedenkapp et al., 2021; Sharma et al., 2017; Yu et al., 2021).

For instance, FiGaR (Sharma et al., 2017) shifts the problem of macro-action learning to predicting the number of steps that the outputted action can be repeated. TempoRL (Biedenkapp et al., 2021) improved upon FiGaR by conditioning the number of repetitions on the selected actions. However, none of these algorithms can scale to continuous control tasks with multiple action dimensions, as action repetition forces all actuators and joints to be synchronized in their repetitions, leading to poor performance for longer action sequences.

TLA (Patel et al., 2024) has recently shown an enhancement of TempoRL through the implementation of two hierarchical policies functioning at varying timesteps, coordinated by a third policy. Although TLA exhibits commendable results in environments characterized by a single action dimension, its advantages are constrained in multi-dimensional environments. This limitation arises as extended timesteps necessitate synchronization across all degrees of freedom, thereby diminishing

performance during prolonged timesteps. In contrast, SRL is capable of executing distinct actions at each timestep without an increase in decision count.

## 3.4 TEMPORALLY CORRELATED EXPLORATION

Recent advancements in reinforcement learning have extended the concepts of macro-actions and action-repetition to improve exploration by incorporating temporally correlated exploration, where successive actions during exploration exhibit temporal dependencies. For instance, Dabney et al. (2021) proposed temporally extended $\epsilon$-greedy exploration, which involves repeating actions for random durations during exploration. Building on this foundation, subsequent works have investigated approaches such as state-dependent exploration Raffin et al. (2022), episodic reinforcement learning Li et al. (2024), and temporally correlated latent noise Chiappa et al. (2024) to enhance exploration efficiency and improve the smoothness of resulting policies. However, these methods are limited in their adaptability to challenges such as observational dropout, low decision or observational frequency, as the trained policy requires state input at each timestep. To address long-horizon temporally correlated exploration, Zhang et al. (2022) introduced the Generative Planning Method (GPM), which employs a recurrent actor network similar to the architecture used in SRL to generate sequences of actions from a single state. We provide an empirical comparison to GPM in Section 5.

## 4 SEQUENCE REINFORCEMENT LEARNING

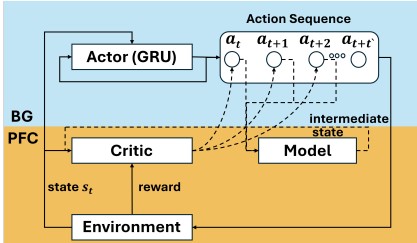

Figure 1: The Sequence Reinforcement Learning (SRL) model. The SRL takes inspiration from the function of the basal ganglia (BG) (Top/Blue) and the prefrontal cortex (PFC) (Bottom/Yellow). We train an actor with a gated recurrent unit that can produce sequences of arbitrary lengths given a single state. This is achieved by utilizing a critic and a model that acts at a finer temporal resolution during training/replay to provide an error signal to each primitive action of the action sequence.

We introduce a novel reinforcement learning model capable of learning sequences of actions (macro-actions) by replaying memories at a finer temporal resolution than the action generation, utilizing a model of the environment during training. We provide the neural basis for our algorithm in the Appendix (A.9)

### COMPONENTS

The Sequence Reinforcement Learning (SRL) algorithm learns to plan "in-the-mind" using a model during training, allowing the learned action-sequences to be executed without the need for model-based online planning. This is achieved using an actor-critic setting where the actor and critic operate at different frequencies, representing the observation/computation and actuation frequencies, respectively. Essentially, the critic is only used during training/replay and can operate at any temporal resolution, while the actor is constrained to the temporal resolution of the slowest component in the sensing-compute-actuation loop. Denoting the actor's timestep as $t'$ and the critic's timestep as $t$, our algorithm includes three components:

$$
\begin{aligned}
\text{Model} &: s_{t+1} = \mathbf{m}_\phi(s_t, a_t) \\
\text{Critic} &: q_t = \mathbf{q}_\psi(s_t, a_t) \\
\text{Actor} &: m_{t':t'+J-1} = a_{t'}, a_{t'+t}, a_{t'+2t}.. \sim \pi_\omega(s_{t'})
\end{aligned}
\tag{1}
$$

We denote individual actions in the action sequence generated by the actor using the notation $\pi_\omega(s_{t'})_t$

We denote individual actions in the action sequence $m_{t':t'+J-1} = a_{t'}, a_{t'+t}, a_{t'+2t}..$ generated by the actor using the notation $\pi_\omega(s_{t'})_t$ to represent the action $a_{t'+t}$.

1. **Model**: Learns the dynamics of the environment, predicting the next state $s_{t+1}$ given the current state $s_t$ and primitive action $a_t$.

2. **Critic**: Takes the same input as the model but predicts the Q-value of the state-action pair.

3. **Actor**: Produces a sequence of actions given an observation at time $t'$. Observations from the environment can occur at any timestep $t$ or $t'$, where we assume $t' > t$. Specifically, in our algorithm, $t' = Jt$ where $J > 1; J \in \mathbb{Z}$.

Each component of our algorithm is trained in parallel, demonstrating competitive learning speeds.

We follow the Soft-Actor-Critic (SAC) algorithm (Haarnoja et al., 2019) for learning the actor-critic. Exploration and uncertainty are critical factors heavily influenced by timestep size and planning horizon. Many model-free algorithms like DDPG (Lillicrap et al., 2019) and TD3 (Fujimoto et al., 2018) explore by adding random noise to each action during training. However, planning a sequence of actions over a longer timestep can result in additive noise, leading to poor performance during training and exploration if the noise parameter is not tuned properly. The SAC algorithm addresses this by automatically maximizing the entropy while also maximizing the expected return, allowing our algorithm to automatically tune its exploration based on the selected sequence length parameter ($J$).

LEARNING THE MODEL

The model is trained to minimize the Mean Squared Error of the predicted states. For a trajectory $\tau = (s_t, a_t, s_{t+1})$ drawn from the replay buffer $\mathcal{D}$, the predicted state is taken from $\tilde{s}_{t+1} \sim \mathbf{m}_\phi(s_t, a_t)$. The loss function is:

$$\mathcal{L}_\phi = \mathbb{E}_{\tau \sim \mathcal{D}}(\tilde{s}_{t+1} - s_{t+1})^2 \tag{2}$$

For this work, the model is a feed-forward neural network with two hidden layers. In addition to the current model $\mathbf{m}_\phi$, we also maintain a target model $\mathbf{m}_{\phi^-}$ that is the exponential moving average of the current model.

LEARNING THE CRITIC

The critic is trained to predict the Q-value of a given state-action pair $\tilde{q}_t = \mathbf{q}_\psi(s_t, a_t)$ using the target value from the modified Bellman equation:

$$\hat{q}_t = r_t + \gamma \mathbb{E}_{a_{t+1} \sim \pi_\omega(s_{t+1})_0}[\mathbf{q}_{\psi^-}(s_{t+1}, a_{t+1}) - \alpha \log \pi_\omega(a_{t+1}|s_{t+1})] \tag{3}$$

Here, $\mathbf{q}_{\psi^-}$ is the target critic, which is the exponential moving average of the critic and $\alpha$ is the temperature parameter that controls the relative importance of the entropy term. Following the SAC algorithm, we train two critics and use the minimum of the two $\mathbf{q}_{\psi^-}$ values to train the current critics. The loss function is:

$$\mathcal{L}_\psi = \mathbb{E}_{\tau \sim \mathcal{D}}[(\tilde{q}_{tk} - \hat{q}_t)^2] \forall k \in 1, 2 \tag{4}$$

Both critics are feed-forward neural networks with two hidden layers. It should be noted that while the actor utilizes the model during training, the critic does not train on any data generated by the model, thus the critic training is model-free and grounded in the real environment states.

LEARNING THE POLICY

The SRL policy utilizes two hidden layers followed by a Gated-Recurrent-Unit (GRU) (Cho et al., 2014) that takes as input the previous action in the action sequence, followed by two linear layers that output the mean and standard deviation of the Gaussian distribution of the action. This design allows the policy to produce action sequences of arbitrary length given a single state and the last action.

A naive approach to training a sequence of actions would be to augment the action space to include all possible actions of the sequence length. However, this quickly leads to the curse of dimensionality, as each sequence is considered a unique action, dramatically increasing the policy's complexity.

Additionally, such an approach ignores the temporal information of the action sequence and faces the difficult problem of credit assignment, with only a single scalar reward for the entire action sequence.

To address these problems, we use different temporal scales for the actor and critic. The critic assigns value to each primitive action of the action sequence, bypassing the credit assignment problem caused by the single scalar reward. However, using collected state-action transitions to train the action sequence is impractical, as changing the first action in the sequence would render all future states inaccurate. Thus, the model populates intermediate states, which the critic then uses to assign value to each primitive action in the sequence.

Therefore, given a trajectory $\tau = (a_{t-1}, s_t, a_t, s_{t+1})$, we first produce the $J$-step action sequence using the policy: $\tilde{m}_{t:t+J-1} \sim \pi_\omega(s_t)$. We then iteratively apply the target model to get the intermediate states $\tilde{s}_{t+1:t+J-1}$. Finally, we use the critic to calculate the loss for the actor as follows:

$$\mathcal{L}_\omega = \mathbb{E}_{\tau \sim \mathcal{D}} \left[ \alpha \log \pi_\omega(\tilde{a}_t | s_t) - \mathbf{q}_\psi(s_t, \tilde{a}_t) + \sum_{j=1}^{J-1} \alpha \log \pi_\omega(\tilde{a}_{t+j} | \tilde{s}_{t+j}) - \mathbf{q}_\psi(\tilde{s}_{t+j}, \tilde{a}_{t+j}) \right] \quad (5)$$

## 5 EXPERIMENTS

### OVERVIEW

We evaluate our SRL approach on 11 continuous control tasks, comparing it against SAC (Haarnoja et al., 2019) and GPM (Zhang et al., 2022). We utilize the OpenAI Gym (Brockman et al., 2016) implementation of the MuJoCo environments (Todorov et al., 2012).

### EXPERIMENTAL SETUP

We train SRL with four different action sequence lengths (ASL), $J = 2, 4, 8, 16$, referred to as SRL-$J$. During training, SRL is evaluated based on its $J$ value, processing states only after every $J$ actions. All hyperparameters are identical between SRL and SAC, except for the actor update frequency: SRL updates the actor every 4 steps, while SAC updates every step. Thus, SAC has four more actor update steps compared to SRL. Additionally, SRL learns a model in parallel with the actor and critic. Additionally, we also train SAC at different step sizes that correspond to SRL, forming SAC-$J$ where $J = 1, 2, 4, 8, 16$. Note that we do not provide SRL-1 since for sequences of length 1, SRL is the same algorithm as SAC.

We present the learning curves of SRL and SAC across 11 continuous control tasks in the appendix. We find that on all environments except Swimmer, SAC-1 demonstrates optimal performance and often significantly outperforms the longer timesteps. Thus, the default environments are picked to maximize performance under the standard RL setting where the observation, decision, and the action frequency are the same. It should be noted that the learning curves presented for SRL-$J$ and SAC-$J$ take in states every $J$ steps.

### FREQUENCY-AVERAGED SCORE

Transitioning from simulation to real-world implementation (Sim2Real) in control systems is challenging because deployment introduces computational stochasticity, leading to variable sensor sampling rates (throughput) and inconsistent end-to-end delays from sensing to actuation (Sandha et al., 2021). This gap is not captured by the mean reward or return that is the norm in current RL literature. To address this, we introduce Frequency-Averaged Score (FAS) that is the normalized area under the curve (AUC) of the performance vs. decision frequency plot. We provide plots for all environments in the Appendix. We note that this experimental setup is similar to the challenge 7 introduced in by Dulac-Arnold et al. (2021) and SRL addresses the challenge of low throughput that is introduced in that work. The FAS captures the overall performance of the policy at different decision frequencies, timesteps or macro-action lengths. A High FAS indicates that the policy performance generalizes across decision frequencies, observation frequencies and timestep sizes.

Tables 2 and 3 present the Frequency Averaged Score (FAS) for SAC and SRL across varying action sequence lengths. Overall, SRL-16 demonstrates strong and consistent performance

| Environment | SAC-1 | SAC-2 | SAC-4 | SAC-8 | SAC-16 |
|---|---|---|---|---|---|
| Pendulum | 0.44 ± 0.03 | 0.42 ± 0.03 | **0.50** ± 0.03 | 0.49 ± 0.04 | 0.33 ± 0.05 |
| Lunar Lander | 0.20 ± 0.02 | 0.23 ± 0.02 | 0.33 ± 0.02 | 0.45 ± 0.03 | **0.56** ± 0.09 |
| Hopper | 0.07 ± 0.01 | 0.09 ± 0.01 | 0.14 ± 0.03 | 0.14 ± 0.04 | **0.26** ± 0.08 |
| Walker2d | 0.07 ± 0.01 | 0.08 ± 0.03 | 0.14 ± 0.04 | **0.23** ± 0.07 | 0.15 ± 0.04 |
| Ant | -0.05 ± 0.04 | 0.11 ± 0.01 | 0.16 ± 0.02 | **0.16** ± 0.01 | 0.13 ± 0.01 |
| HalfCheetah | 0.01 ± 0.01 | **0.04** ± 0.01 | 0.03 ± 0.00 | 0.02 ± 0.01 | 0.01 ± 0.01 |
| Humanoid | 0.06 ± 0.01 | 0.06 ± 0.01 | 0.08 ± 0.03 | 0.17 ± 0.02 | **0.18** ± 0.04 |
| InvertedPendulum | 0.05 ± 0.02 | 0.07 ± 0.00 | 0.14 ± 0.00 | 0.31 ± 0.02 | **0.34** ± 0.20 |
| InvertedDPendulum | 0.02 ± 0.00 | 0.07 ± 0.00 | **0.09** ± 0.01 | 0.01 ± 0.00 | 0.01 ± 0.00 |
| Reacher | 0.65 ± 0.07 | 0.78 ± 0.01 | 0.84 ± 0.03 | 0.86 ± 0.02 | **0.87** ± 0.02 |
| Swimmer | 0.08 ± 0.02 | 0.28 ± 0.04 | 0.46 ± 0.03 | 0.53 ± 0.03 | **0.54** ± 0.06 |

Table 2: Mean Frequency-Averaged Score (FAS) and standard deviation for different environments for SAC-$J$ configurations ($J = 1, 2, 4, 8, 16$. $J$ is the action sequence length during training). Each value is averaged over 5 trials (rounded to two decimals, highest value highlighted).

| Environment | SRL-2 | SRL-4 | SRL-8 | SRL-16 |
|---|---|---|---|---|
| Pendulum | 0.49 ± 0.04 | 0.68 ± 0.02 | 0.78 ± 0.04 | **0.88** ± 0.02 |
| Lunar Lander | 0.14 ± 0.06 | 0.52 ± 0.03 | 0.73 ± 0.04 | **0.84** ± 0.03 |
| Hopper | 0.10 ± 0.02 | 0.23 ± 0.03 | 0.42 ± 0.04 | **0.57** ± 0.02 |
| Walker2d | 0.12 ± 0.03 | 0.25 ± 0.06 | **0.28** ± 0.06 | 0.24 ± 0.11 |
| Ant | 0.04 ± 0.01 | 0.29 ± 0.09 | 0.45 ± 0.14 | **0.54** ± 0.13 |
| HalfCheetah | 0.06 ± 0.01 | 0.13 ± 0.02 | 0.22 ± 0.01 | **0.28** ± 0.01 |
| Humanoid | 0.07 ± 0.00 | 0.18 ± 0.02 | 0.37 ± 0.04 | **0.46** ± 0.04 |
| InvertedPendulum | 0.09 ± 0.03 | 0.16 ± 0.03 | 0.27 ± 0.02 | **0.44** ± 0.04 |
| InvertedDPendulum | 0.07 ± 0.00 | **0.13** ± 0.02 | 0.03 ± 0.02 | 0.02 ± 0.00 |
| Reacher | 0.90 ± 0.01 | 0.93 ± 0.00 | 0.95 ± 0.00 | **0.96** ± 0.00 |
| Swimmer | 0.32 ± 0.05 | 0.38 ± 0.17 | 0.31 ± 0.02 | **0.42** ± 0.15 |

Table 3: Mean Frequency-Averaged Score (FAS) and standard deviation for different environments for SRL-$J$ configurations ($J = 2, 4, 8, 16$. $J$ is the action sequence length during training). Each value is averaged over 5 trials (rounded to two decimals, highest value highlighted).

across most environments and a wide range of frequencies. However, in the Walker2d-v2 and InvertedDoublePendulum-v2 environments, SRL faces challenges when learning longer action sequences. We hypothesize that these difficulties stem from higher modeling errors in these environments. Future work aimed at improving environmental models could potentially address these issues.

SAC, in contrast, performed poorly across all environments, highlighting the limitations of traditional RL methods in adapting to changes in frequency. Although training SAC with larger timesteps ($J$) improves FAS, this approach compromises performance at shorter timesteps, ultimately reducing the overall score (see Appendix Fig. 5).

An exception to this trend is the Swimmer environment, where SAC benefits from improved exploration due to extended actions. SRL, which does not use action repetition, does not perform as well in this specific case. However, this limitation could be addressed by incorporating action repetition or action correlation during exploration—an enhancement that lies beyond the scope of the current work.

In order to further validate the utility of FAS, we test all the policies (SAC and SRL-$J$) in a stochastic timestep environment. The timestep (time until next input) is randomly chosen from a uniform distribution of integers in [1,16] after each decision. This is a more realistic setting as it tests the performance of the policy when the frequency is not constant. Each policy is evaluated over 10 episodes with stochastic timesteps.

In all tested environments, except for the Inverted Double Pendulum, there is a strong Pearson correlation coefficient (greater than or equal to 0.82) between FAS and performance in stochastic

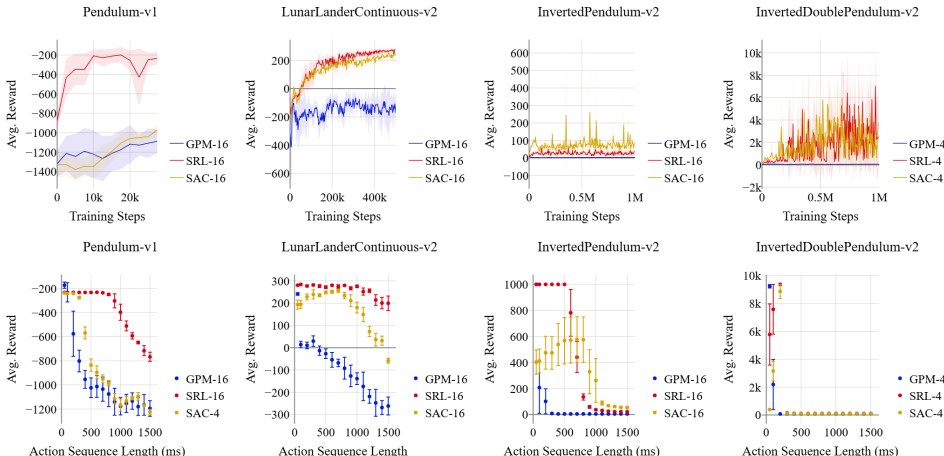

Figure 2: Comparison of SAC and SRL to GPM. Top: Learning curves. Bottom: Performance of the trained policies at different action sequence lengths. The action sequences for SRL and GPM are generated using the recurrent actor while SAC utilizes action repetition. GPM achieves FAS of 0.41, 0.04, 0.04, 0.04 on the environments from left to right respectively.

conditions. This high correlation confirms the effectiveness of FAS as a metric for measuring a policy's generalized performance across various timesteps and frequencies. The Inverted Double Pendulum, however, presents a unique challenge due to its requirement for high precision at low decision frequencies, leading to significantly lower FAS scores for all algorithms and thus it is an outlier. Comprehensive plots for all nine environments are included in the appendix (Fig. 7).

### COMPARISON TO GENERATIVE PLANNING METHOD

The Generative Planning Method (GPM) (Zhang et al., 2022) uses a recurrent actor, like SRL, to generate actions for improved exploration. Originally designed for a different context and evaluated in the standard RL setting, GPM optimizes plan actions to maximize Q-value, potentially exceeding SAC in FAS score. We compare SRL and GPM in four environments to test this.

In the original work, GPM was trained with a plan length of 3, similar to the $J$ parameter in our study. Though shorter plans may restrict generalization to longer sequences, GPM is robust to plan length variations. For fair comparison, we use the best-performing $J$ values for SRL in each environment.

Figure 2 shows the learning curves and FAS evaluation plots for GPM compared to SAC and SRL. While GPM generates a plan by optimizing a sequence of actions, it achieves optimal performance only at sequence lengths of one. As a result, its FAS score is even lower than that of SAC-$J$.

Notably, on the InvertedDoublePendulum-v2 environment, both SAC and SRL exhibit high performance at action sequence lengths (ASL) of 4, which aligns with their training at $J = 4$. However, their performance decreases at shorter ASLs. In contrast, GPM shows a similar FAS profile to SAC-1, indicating that its performance does not generalize well to longer action sequences.

### COMPARISON TO MODEL-BASED ONLINE PLANNING

Model-based online planning is another approach that allows the RL agent to reduce its observational frequency. However, it often requires a highly accurate model of the environment and incurs increased model complexity due to the use of the model during control.

Since SRL incorporates a model of the environment that is learned in parallel, we compare the performance of the SRL actor utilizing the actor-generated action sequences against model-based online planning, where the actor produces only a single action between each simulated state.

Table 4 compares the FAS score SRL to online planning using the same model in online planning versus the action sequences generated by the SRL policy. We see that SRL can learn action

| Environment | SRL | Online Planning | State Space | Action Space |
|---|---|---|---|---|
| Lunar Lander | **0.84 ± 0.03** | 0.79 ± 0.08 | 8 | 2 |
| Hopper | 0.57 ± 0.02 | **0.59 ± 0.19** | 11 | 3 |
| Walker2d | **0.28 ± 0.06** | 0.20 ± 0.05 | 17 | 6 |
| Ant | **0.54 ± 0.13** | 0.34 ± 0.08 | 27 | 8 |
| HalfCheetah | **0.28 ± 0.01** | 0.19 ± 0.02 | 17 | 6 |
| Humanoid | **0.46 ± 0.04** | 0.18 ± 0.03 | 376 | 17 |
| InvPendulum | 0.44 ± 0.04 | **0.63 ± 0.10** | 4 | 1 |
| InvDPendulum | **0.13 ± 0.02** | 0.10 ± 0.07 | 11 | 1 |
| Reacher | **0.96 ± 0.00** | 0.95 ± 0.00 | 11 | 2 |
| Swimmer | 0.42 ± 0.15 | **0.43 ± 0.14** | 8 | 2 |

Table 4: Comparison of the FAS of SRL and corresponding model-based online planning policies across different environments.

sequences and is competitive to model-based online planning. Notably, SRL performs better in environments with larger action and state space dimensions. Such environments are harder to model. Thus, SRL can leverage inaccurate models to learn accurate action sequences, further reducing the required computational complexity during training. We hypothesize that this superior performance is due to the fact that the actor learns a $J$-step action sequence concurrently, while online planning only produces one action at a time. Consequently, SRL is able to learn and produce long, coherent action sequences, whereas single-step predictions tend to drift, similar to the 'hallucination' phenomenon observed in transformer-based language models.

## 6   DISCUSSION AND FUTURE WORK

SRL bridges the gap between RL and real-world applications by enabling robust control at low decision frequencies. Its ability to learn long action sequences expands the potential for deploying RL in resource-constrained environments, such as robotics and autonomous systems. Additionally, it shows promise for applications where obtaining observations is costly, such as in medical diagnostics and treatment planning. Future work will explore hierarchical policies and biologically inspired attention mechanisms.

The current RL framework encourages synchrony between the environment and the components of the agent. However, the brain utilizes components that act at different frequencies and yet is capable of robust and accurate control. SRL provides an approach to reconcile this difference between neuroscience and RL, while remaining competitive on current RL benchmarks. SRL offers substantial benefits over traditional RL algorithms, particularly in the context of autonomous agents in constrained settings. By enabling operation at slower observational frequencies and providing a gradual decay in performance with reduced input frequency, SRL addresses critical issues related to sensor failure and occlusion, and energy consumption. Additionally, SRL generates long sequences of actions from a single state, which can enhance the explainability of the policy and provide opportunities to override the policy early in case of safety concerns. SRL also learns a latent representation of the action sequence, which could be used in the future to interface with large language models for multimodal explainability and even hierarchical reinforcement learning and transfer learning.

## 7   CONCLUSION

In this paper, we introduced Sequence Reinforcement Learning (SRL): a model-based action sequence learning algorithm for model-free control. We demonstrated the improvement of SRL over the existing framework by testing it over various control frequencies. Furthermore, we introduce the Frequency-Averaged-Score (FAS) metric to measure the robustness of a policy across different frequencies. Our work is the first to achieve competitive results on continuous control environments at low control frequencies and serves as a benchmark for future work in this direction. Finally, we demonstrated directions for future work, including comparison to model-based planning, generative replay, and connections to neuroscience.

ACKNOWLEDGMENTS

We would like to thank Dr. Terrence Sejnowski for his valuable discussions, insightful feedback, and guidance throughout this work. His expertise and support have been instrumental in refining the ideas presented in this paper.

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

# A  APPENDIX

## TABLE OF CONTENTS

## A.1  SRL ALGORITHM

---

**Algorithm 1:** Sequence Reinforcement Learning

---

**Input:** $\phi, \psi_1, \psi_2, \omega$. Initial parameters

1 $\bar{\phi} \leftarrow \phi, \bar{\psi}_1 \leftarrow \psi_1, \bar{\psi}_2 \leftarrow \psi_2$ ;          // Initialize target network weights

2 $D \leftarrow \emptyset$ ;                 // Initialize an empty replay pool

3 **for** *each iteration* **do**

4      $\{a_t, a_{t+1}, \ldots, a_{t+J-1}\} \sim \pi_\omega(\{a_t, a_{t+1}, \ldots, a_{t+J-1}\}|s_t)$ ;    // Sample action sequence from the policy

5      **for** *each action $a_t$ in the sequence* **do**

6          $s_{t+1} \sim p(s_{t+1}|s_t, a_t)$ ;  // Sample transition from the environment

7          $D \leftarrow D \cup \{(s_t, a_t, r(s_t, a_t), s_{t+1})\}$ ; // Store transition in the replay pool

8      **end**

9      **for** *each gradient step* **do**

10          $\phi \leftarrow \phi - \lambda_\mathbf{m} \nabla_\phi \mathcal{L}_\phi$ ;         // Update the model parameters

11          **for** $i \in \{1, 2\}$ **do**

12             $\psi_i \leftarrow \psi_i - \lambda_Q \nabla_{\psi_i} \mathcal{L}_{\psi_i}$ ;    // Update the Q-function parameters

13          **end**

14          $\{a_t, a_{t+1}, \ldots, a_{t+J-1}\} \sim \pi_\omega(\{a_t, a_{t+1}, \ldots, a_{t+J-1}\}|s_t)$ ;   // Sample action sequence from the policy

15          **if** *iteration mod actor_update_frequency == 0* **then**

16             **for** $j \in \{1, \ldots, J\}$ **do**

17                 $s_{j+1} \sim \mathbf{m}_{\bar{\phi}}(s_{j+1}|s_j, a_j)$ ;        // Sample transition from the target model

18             **end**

19             $\phi \leftarrow \omega - \lambda_\pi \nabla_\omega L_\omega$ ;         // Update policy weights

20          **end**

21          $\alpha \leftarrow \alpha - \lambda \nabla_{\hat{\alpha}} \mathcal{L}(\alpha)$ ;            // Adjust temperature

22          **for** $i \in \{1, 2\}$ **do**

23             $\bar{\psi}_i \leftarrow \tau \psi_i + (1 - \tau)\bar{\psi}_i$ ;      // Update target network weights

24          **end**

25          $\bar{\phi} \leftarrow \tau \phi + (1 - \tau)\bar{\phi}$ ;         // Update target model weights

26      **end**

27 **end**

**Output:** $\phi, \psi_1, \psi_2, \omega$;                // Optimized parameters

---

## A.2 HYPERPARAMETERS

The table below lists the hyperparameters that are common between every environment used for all our experiments for the SAC and SRL algorithms:

| Hyperparameter | Value | description |
|---|---|---|
| Hidden Layer Size | 256 | Size of the hidden layers in the feed forward networks of Actor, Critic, Model and Encoder networks |
| Updates per step | 1 | Number of learning updates per one step in the environment |
| Target Update Interval | 1 | Inverval between each target update |
| $\gamma$ | 0.99 | Discount Factor |
| $\tau$ | 0.005 | Update rate for the target networks (Critic and Model) |
| Learning Rate | 0.0003 | Learning rate for all neural networks |
| Replay Buffer Size | $10^6$ | Size of the replay buffer |
| Batch Size | 256 | Batch size for learning |
| Start Time-steps | 10000 | Initial number of steps where random policy is followed |

Table 5: List of Common hyperparameters

| Environment | max Timestep | Eval frequency |
|---|---|---|
| LunarLanderContinuous-v2 | 500000 | 2500 |
| Hopper-v2 | 1000000 | 5000 |
| Walker2d-v2 | 1000000 | 5000 |
| Ant-v2 | 5000000 | 5000 |
| HalfCheetah-v2 | 5000000 | 5000 |
| Humanoid-v2 | 10000000 | 5000 |

Table 6: List of environment-specific hyperparameters

## A.3 IMPLEMENTATION DETAILS

Due to its added complexity during training, SRL requires longer wall clock time for training when compared to SAC. We performed a minimal hyperparameter search over the actor update frequency parameter on the Hopper environment (tested values: 1, 2, 4, 8, 16). All the other hyperparamters were picked to be equal to the SAC implementation. We also did not perform a hyerparameter search over the size of GRU for the actor. It was picked to have the same size as the hidden layers of the feed forward network of the actor in SAC. The neural network for the model was also picked to have the same architecture as the actor from SAC, thus it has two hidden layers with 256 neurons. Similarly the encoder for the latent SRL implementation was also picked to have the same architecture. For the latent SRL implementation we also add an additional replay buffer to store transitions of length 5, to implement the temporal consistency training for the model. This was done for simplicity of the implementation, and it can be removed since it is redundant to save memory.

All experiments were performed on a GPU cluster the Nvidia 1080ti GPUs. Each run was performed using a single GPU, utilizing 8 CPU cores of Intel(R) Xeon(R) Silver 4116 (24 core) and 16GB of memory.

We utilize the pytorch implementation of SAC (`https://github.com/denisyarats/pytorch_sac`) (Yarats & Kostrikov, 2020). The official github repository for SRL is: `https://github.com/dee0512/Sequence-Reinforcement-Learning`.

### A.4 PRACTICAL CONSIDERATIONS ON LOW-COMPUTE HARDWARE

In this work, we utilize a GRU for action generation. However, we did not test the performance of other recurrent architectures or transformers. Depending on the hardware constraints and the application, a more complicated or simple architecture could be utilized. Furthermore, we also leave the exploration of actor complexity to generalization to larger action sequences to future work.

Autonomous agents often have observation processing before it is fed into the RL algorithm. It should be noted that observation processing often forms a significant portion of the latency while the recurrent portion of the actor for SRL governs the actuation frequency. Furthermore, as mentioned before, SRL can also inherently handle delays by acting in a predictive manner where the sequence of actions performed in anticipation of the next state that is being processed. Furthermore, in such cases, where there is an overlap between two consecutive action sequences, additional MSE loss can be utilize to align two action sequences. We also leave this exploration to future work.

## A.5 LEARNING CURVES

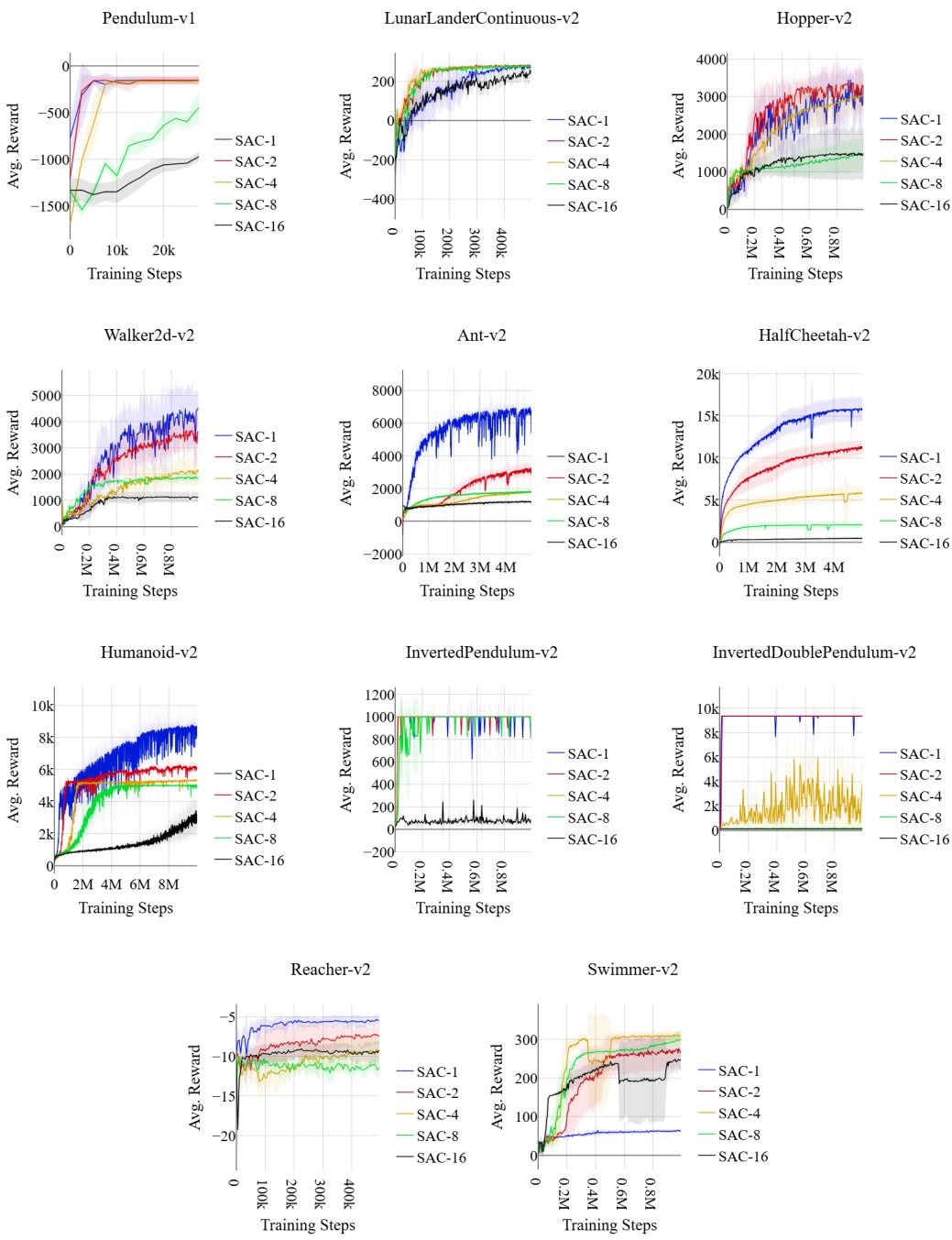

Figure 3: Learning curves for extended action Soft-Actor Critic (SAC-$J$) (Haarnoja et al., 2019) over continuous control tasks. The default timestep $J = 1$ is the optimal for all environments except the swimmer and lunar lander. Larger timesteps support better exploration but also result in worse performance. These results demonstrate that on all environments except swimmer and lunar-lander, the default timestep is picked to optimize for the sweet-spot between better exploration and better performance.

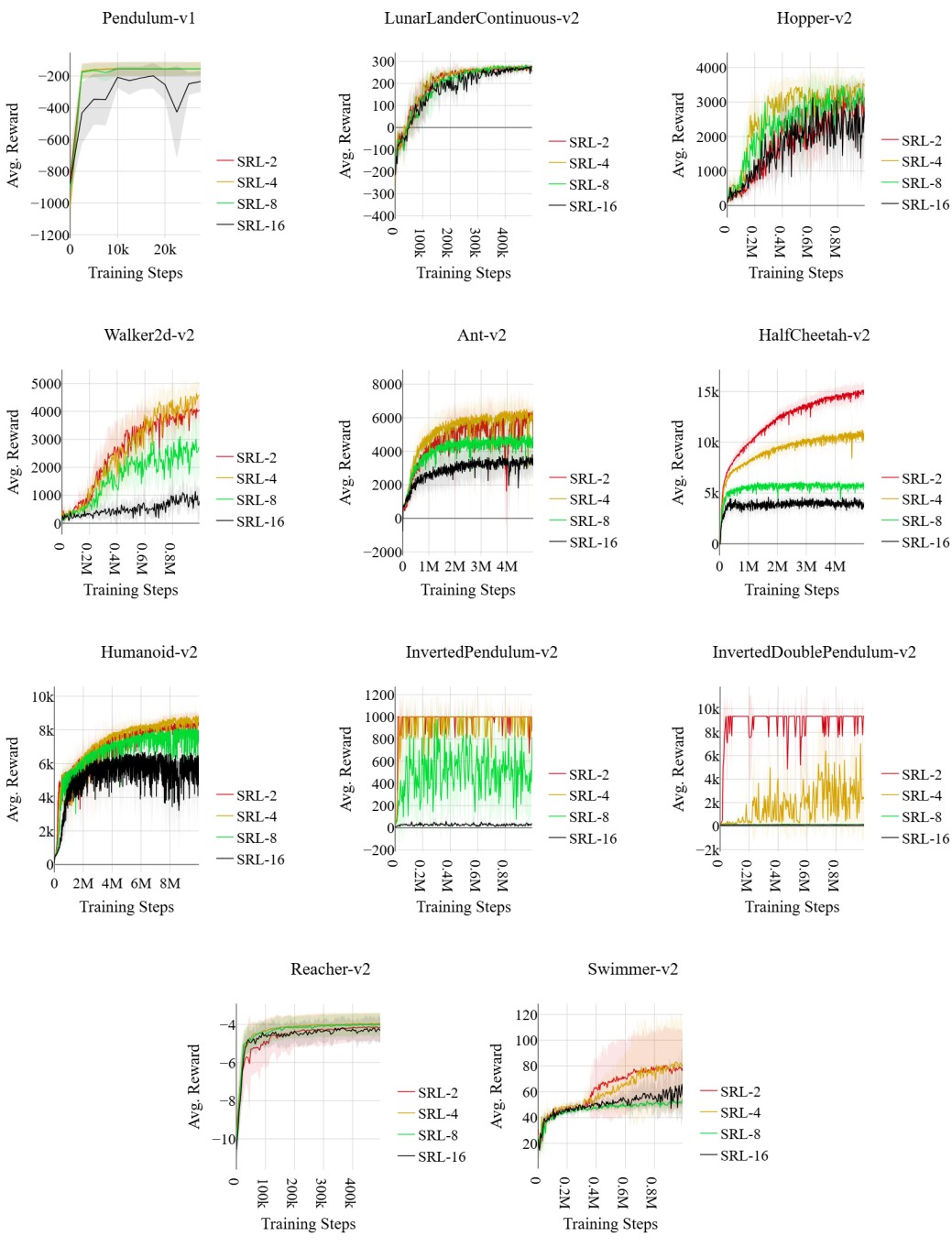

Figure 4: Learning curves of SRL-$J$ (Haarnoja et al., 2019) over continuous control tasks. During evaluation, SRL receives input after $J$ primitive actions. All curves are averaged over 5 trials, with shaded regions representing standard deviation.

## A.6 PLOTS FOR FREQUENCY AVERAGED SCORES

Figure 6 shows the plots for FAS. The ASL of 1 in the figure represents the performance of each policy in the standard reinforcement learning setting. We can see that SRL is competitive with SAC on ASL of 1 on all environments tested. Larger $H$ results in better robustness at longer ASLs but it often comes at the cost of lower performance at shorter ASLs.

Additionally, as the FAS reflects, SRL is also significantly more robust across different frequencies than standard RL (SAC).

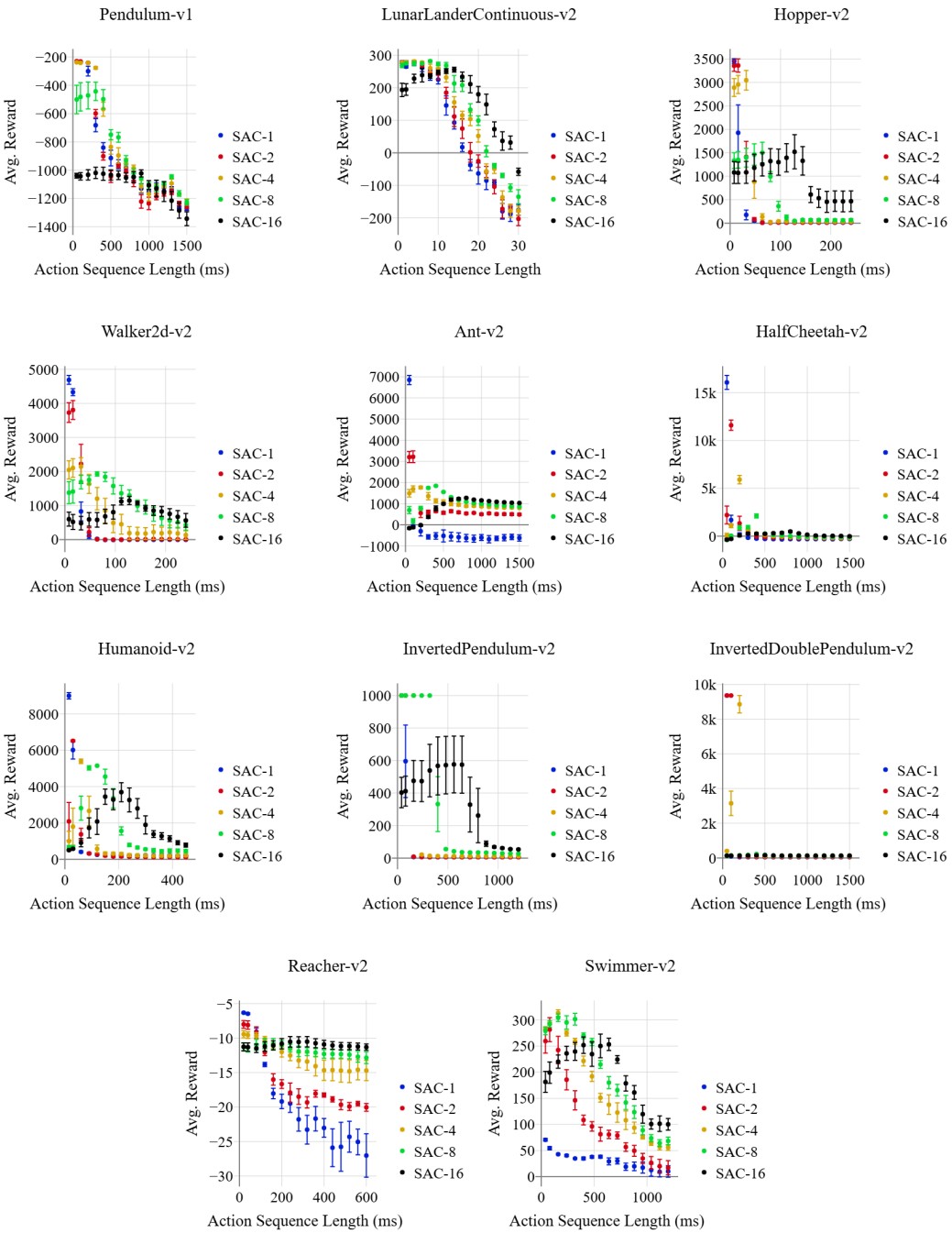

Figure 5: Performance of SAC-*J* at different Action Sequence Lengths (ASL). SAC repeats the same action for the duration. All policies were tested on ASL of 1, 2, 4, 8 ... 30. All markers are averaged over 5 trials, with the error bars representing standard error.

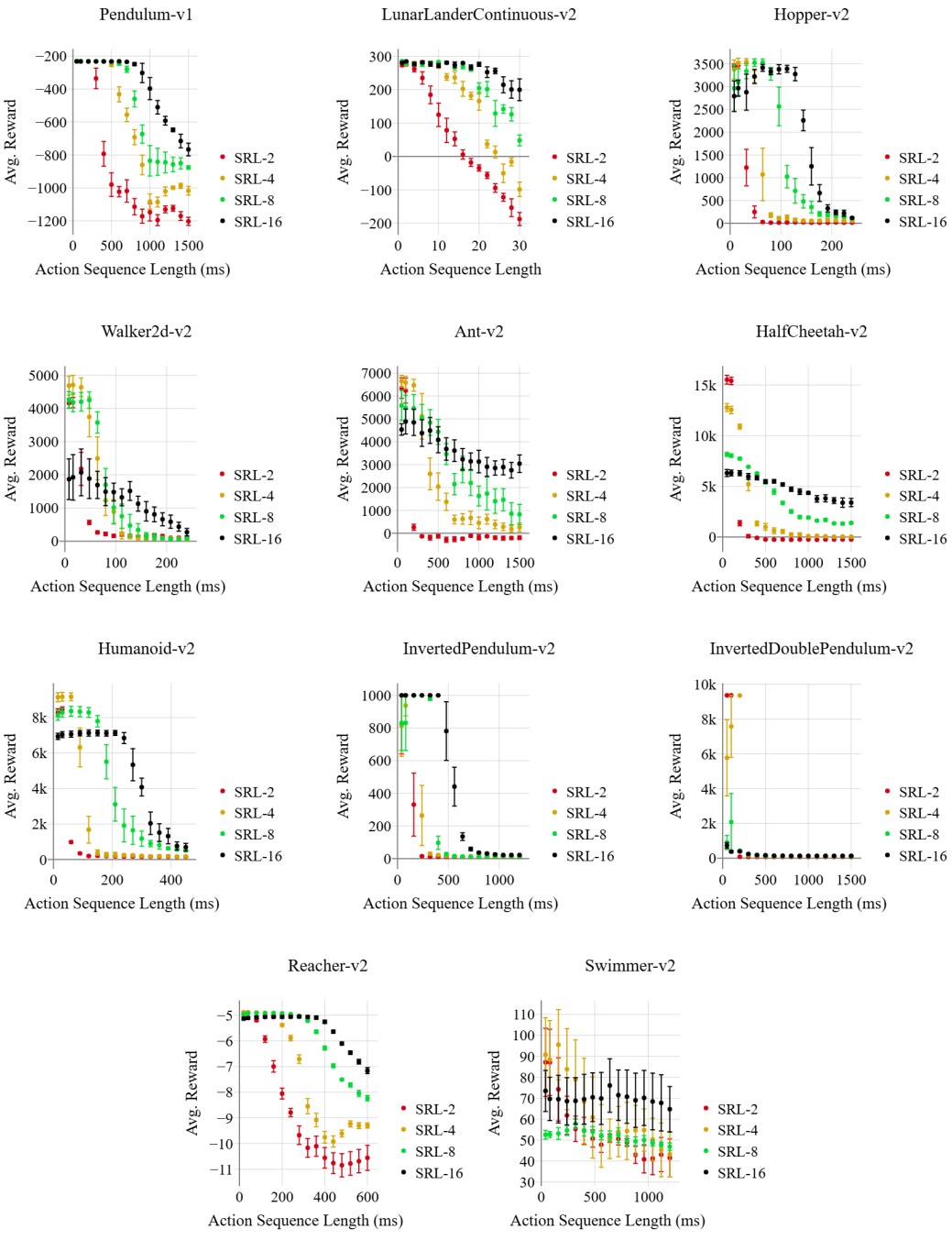

Figure 6: Performance of SRL-$J$ at different Action Sequence Lengths (ASL). All policies were tested on ASL of 1, 2, 4, 8 ... 30. All markers are averaged over 5 trials, with the error bars representing standard error.

## A.7 PLOTS FOR FAS VS. PERFORMANCE FOR STOCHASTIC TIMESTEP

In Figure 7, we present the plots for FAS vs performance for all environments. For all environments except InvertedDoublePendulum-v2, we see a high correlation. InvertedDoublePendulum-v2 is a difficult problem at slow frequency and demonstrates poor performance of less than 200, thus it does not correlate to FAS.

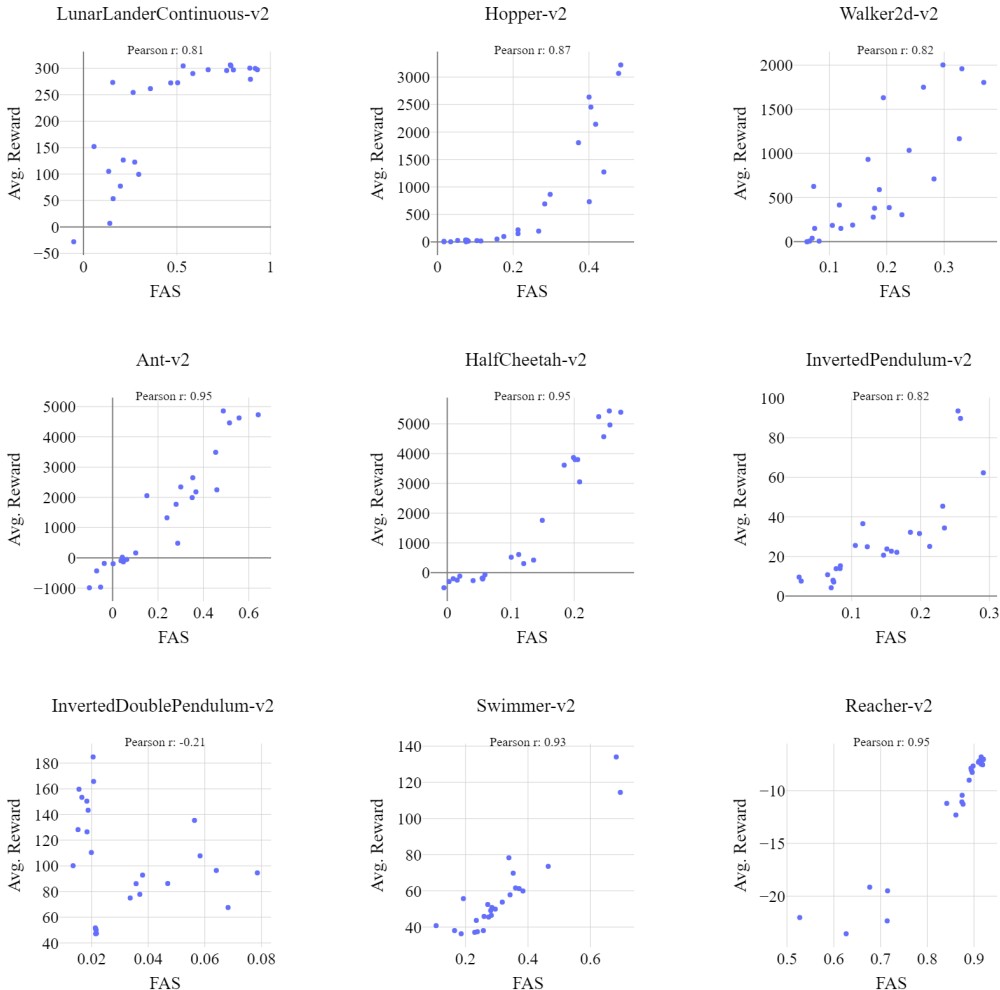

Figure 7: Performance vs. FAS of different policies (SAC, SRL-2, SRL-4, SRL-8, SRL-16). For each algorithm, we test 5 policies over 10 episodes.

## A.8 GENERATIVE REPLAY IN LATENT SPACE

Previous studies have shown that generative replay benefits greatly from latent representations (Van de Ven et al., 2020). Recently, Simplified Temporal Consistency Reinforcement Learning (TCRL) (Zhao et al., 2023) demonstrated that learning a latent state-space improves not only model-based planning but also model-free RL algorithms. Building on this insight, we introduced an encoder to encode the observations in our algorithm.

Following the TCRL implementation, we use two encoders: an online encoder $\mathbf{e}_\theta$ and a target encoder $\mathbf{e}_{\theta^-}$, which is the exponential moving average of the online encoder:

$$\text{Encoder}: e_t = \mathbf{e}_\theta(s_t) \tag{6}$$

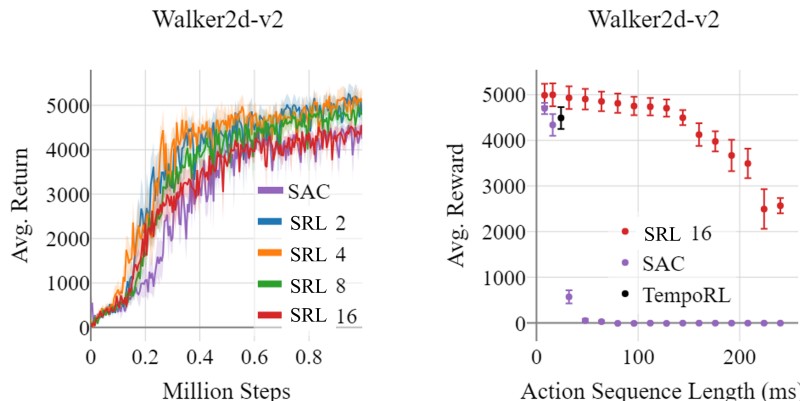

Figure 8: Left: Learning curve of SRL with latent state-space on the Walker2d-v2 environment. Right: Performance of latent SRL-16 on different ASL, compared to SAC and TempoRL. Utilizing a latent representation for state space is especially beneficial for the Walker2d environment so that it outperforms SAC even when training upto sequence lengths of $J = 16$.

Thus, the model predicts the next state in the latent space. Additionally, we introduce multi-step model prediction for temporal consistency. Following the TCRL work, we use a cosine loss for model prediction. The model itself predicts only a single step forward, but we enforce temporal consistency by rolling out the model $H$-steps forward to predict $\tilde{e}_{t+1:t+1+H}$.

Specifically, for an $H$-step trajectory $\tau = (z_t, a_t, z_{t+1})_{t:t+H}$ drawn from the replay buffer $\mathcal{D}$, we use the online encoder to get the first latent state $e_t = \mathbf{e}_\theta(o_t)$. Then conditioning on the sequence of actions $a_{t:t+H}$, the model is applied iteratively to predict the latent states $\tilde{e}_{t+1} = \mathbf{m}_\phi(\tilde{e}_t, a_t)$. Finally, we use the target encoder to calculate the target latent states $\hat{e}_{t+1:t+H+1} = \mathbf{e}_{\theta^-}(o_{t+1:t+1+H})$. The Loss function is defined as:

$$\mathcal{L}_{\theta,\phi} = \mathbb{E}_{\tau \sim \mathcal{D}} \left[ \sum_{h=0}^{H} -\gamma^h \left( \frac{\tilde{e}_{t+h}}{||\tilde{e}_{t+h}||_2} \right)^T \left( \frac{\hat{e}_{t+h}}{||\hat{e}_{t+h}||_2} \right) \right] \tag{7}$$

We set $H = 5$ for our experiments. Both the encoder and the model are feed-forward neural networks with two hidden layers.

We provide preliminary results for the Walker environment. Utilizing the latent space for generative replay significantly improved performance, making it competitive even at 16 steps (128ms) (Figure 8).

We also provide the TempoRL (Biedenkapp et al., 2021) algorithm as a benchmark as it is an algorithm that successfully reduces the number of decisions per episodes. TempoRL is designed to dynamically pick the best frameskip (for performance), therefore we report the avg. action sequence length for TempoRL.

## A.9 NEURAL BASIS FOR SEQUENCE LEARNING

Unlike artificial RL agents, learning in the brain does not stop once an optimal solution has been found. During initial task learning, brain activity increases as expected, reflecting neural recruitment. However, after training and repetition, activity decreases as the brain develops more efficient representations of the action sequence, commonly referred to as muscle memory (Wiestler & Diedrichsen, 2013). This phenomenon is further supported by findings that sequence-specific activity in motor regions evolves based on the amount of training, demonstrating skill-specific efficiency and specialization over time (Wymbs & Grafton, 2015).

The neural basis for action sequence learning involves a sophisticated interconnection of different brain regions, each making a distinct contribution:

1. **Basal ganglia** (BG): Action chunking is a cognitive process by which individual actions are grouped into larger, more manageable units or "chunks," facilitating more efficient storage, retrieval, and execution with reduced cognitive load (Favila et al., 2024). Importantly, this mechanism allows the brain to perform extremely fast and precise sequences of actions that would be impossible if produced individually. The BG plays a crucial role in chunking, encoding entire behavioral action sequences as a single action (Jin et al., 2014; Favila et al., 2024; Jin & Costa, 2015; Berns & Sejnowski, 1996; 1998; Garr, 2019). Dysfunction in the BG is associated with deficits in action sequences and chunking in both animals (Doupe et al., 2005; Jin & Costa, 2010; Matamales et al., 2017) and humans (Phillips et al., 1995; Boyd et al., 2009; Favila et al., 2024). However, the neural basis for the compression of individual actions into sequences remains poorly understood.

2. **Prefrontal cortex** (PFC): The PFC is critical for the active unbinding and dismantling of action sequences to ensure behavioral flexibility and adaptability (Geissler et al., 2021). This suggests that action sequences are not merely learned through repetition; the PFC modifies these sequences based on context and task requirements. Recent research indicates that the PFC supports memory elaboration (Immink et al., 2021) and maintains temporal context information (Shahnazian et al., 2022) in action sequences. The prefrontal cortex receives inputs from the hippocampus.

3. **Hippocampus** (HC) replays neuronal activations of tasks during subsequent sleep at speeds six to seven times faster. This memory replay may explain the compression of slow actions into fast chunks. The replayed trajectories from the HC are consolidated into long-term cortical memories (Zielinski et al., 2020; Malerba et al., 2018). This phenomenon extends to the motor cortex, which replays motor patterns at accelerated speeds during sleep (Rubin et al., 2022).

## A.10 CLARIFICATION FIGURE

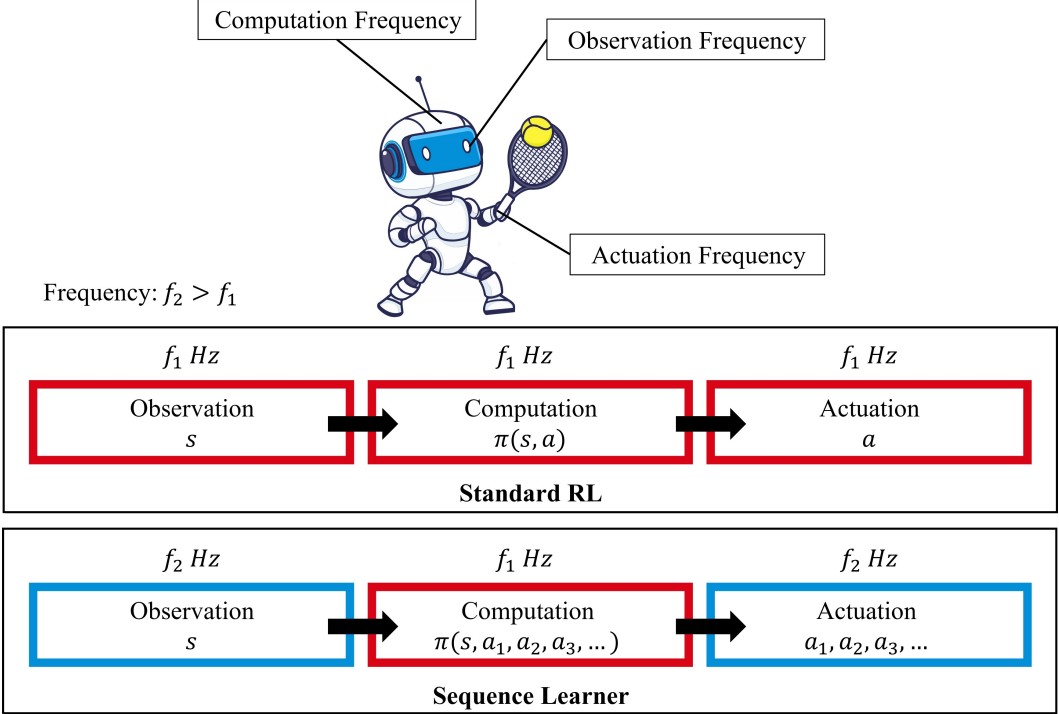

Figure 9: Illustration of the control process in an RL agent, comprising three key components: observation, computation, and actuation. In a standard RL framework, these components typically operate at the same frequency, with each observation leading to a single action after a computation pass. However, the sequence learner can achieve faster actuation by generating multiple primitive actions per observation. It's important to note that during training, the observation frequency must be at least equal to the actuation frequency and, after training, must match the computation frequency.

## A.11 LEARNING CURVES BY J

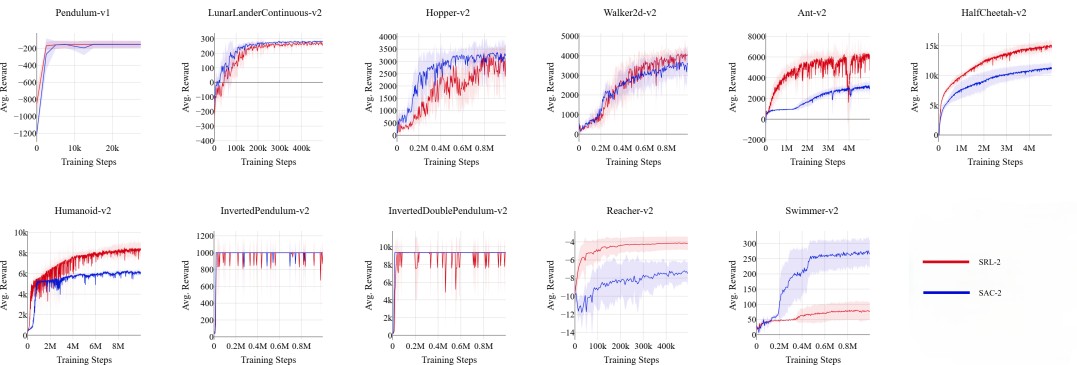

Figure 10: Learning curve of SRL-2 and SAC-2.

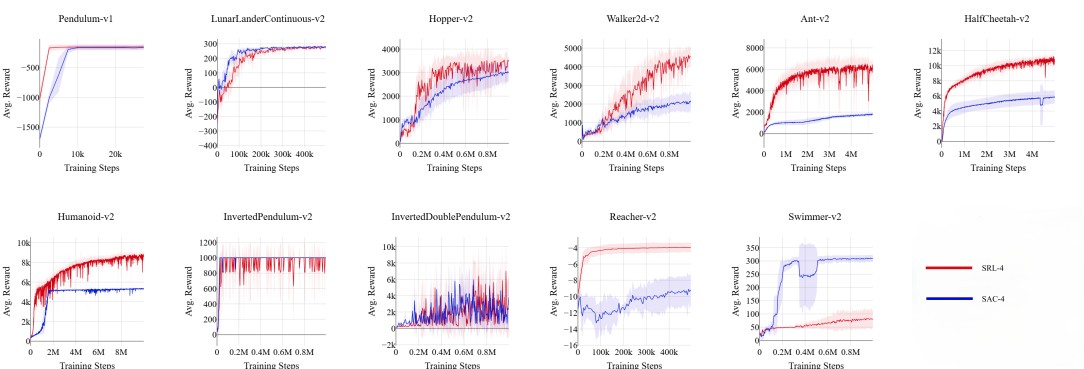

Figure 11: Learning curve of SRL-4 and SAC-4.

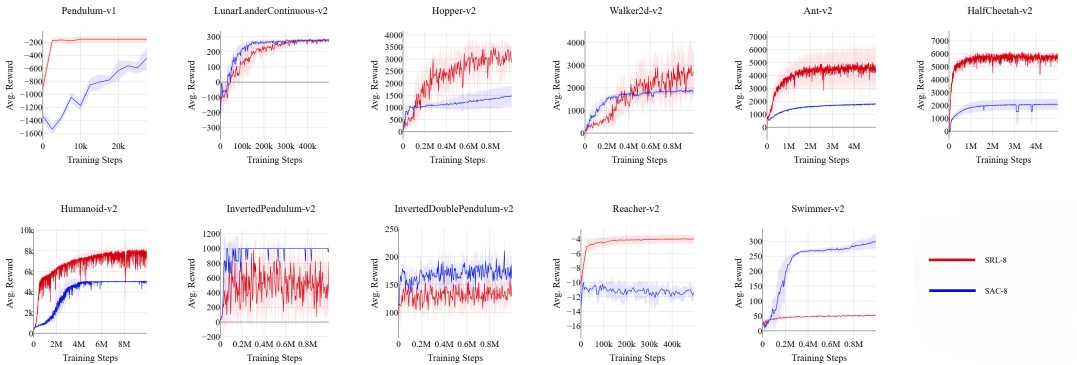

Figure 12: Learning curve of SRL-8 and SAC-8.

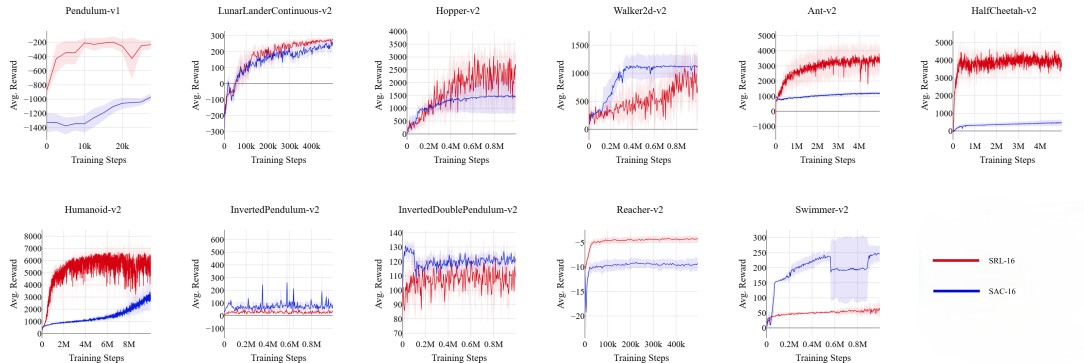

Figure 13: Learning curve of SRL-16 and SAC-16.

## A.12 RANDOMIZED FRAME-SKIPPING

As shown, SAC trained on a constant timestep cannot adapt to different timesteps. For a fairer comparison, we also present results on randomized frame-skipping implemented on SAC during training.

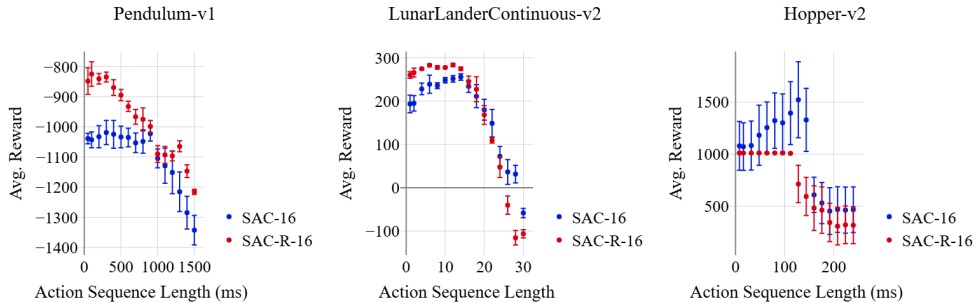

Figure 14: Performance of SAC and randomized SAC (SAC-R).

Figure 14 compares the performance of randomized SAC (SAC-R) to SAC at $J = 16$. Surprisingly, we find that randomized frame-skipping during training improves the performance at shorter action sequence lengths (ASL) for simple environments like pendulum and lunar lander. However, for Hopper, SAC-R performs worse than SAC. This is most probably due to the stochasticity introduced due to the randomized frame-skipping. Even with randomized frame-skipping, SAC fails to achieve performance similar to SRL on simple environments, thus further reinforcing the results presented in this paper.

## A.13 RESULTS FOR TEMPORL ALGORITHM

To further provide provide context for the contribution of this work in comparison to previous work, we provide further comparison to TempoRL (Biedenkapp et al., 2021) and also discuss performance compared to recent work on observational dropout.

TempoRL cannot be adapted to the FAS setting since after each action is picked, it further picks the duration for the amount of time the action will be performed. Yet, since it promotes action repetiton, it results in lower decision frequency and longer action sequence lengths than standard algorithms like TD3 and SAC.

Table 7 demonstrates the results of training TempoRL algorithm on some of the benchmarks presented in this paper. We did a quick hyperparameter search over the max sequence length parameter and pick the highest number over 3 that did not result in a significant drop in performance. We

| Environment | Avg. Reward | Avg. Sequence Length | Max sequence Length |
|---|---|---|---|
| Pendulum | -149.38 ±31.26 | 71.74ms | 6 |
| Hopper | 2607.86 ±342.23 | 22.4ms | 9 |
| Walker2d | 4581.69 ±561.95 | 25.54ms | 7 |
| Ant | 3507.85 ±579.95 | 62.66ms | 3 |
| HalfCheetah | 6627.73 ±2500.77 | 56.20ms | 3 |
| Inv Pendulum | 984.21 ±47.37 | 73.92ms | 10 |
| InvD Pendulum | 9352.61 ±2.2 | 58.76ms | 5 |

Table 7: Results of running TempoRL on Mujoco Tasks. All results are averaged over 10 seeds.

find that while TempoRL achieve optimal performance on environments with single dimensions like pendulums, it demonstrates significant drop in performance on environments with multiple dimensions like Ant and HalfCheetah. Furthermore, on all environments, it maintains a relatively short action sequence length and even though it is given the option of picking long action sequences, it rarely does so. This result further demonstrates the contribution of SRL at maintaining performance at really long sequence lengths in environments with high action dimensions.

