# OpenReview forum: "Overcoming Slow Decision Frequencies in Continuous Control: Model-Based Sequence Reinforcement Learning for Model-Free Control"
_ICLR.cc/2025/Conference — ICLR 2025 Poster_

### Official Review · Reviewer_pfCy · 2024-10-22

**Soundness:** 3
**Presentation:** 3
**Contribution:** 3
**Rating:** 8
**Confidence:** 4

**Summary:**

The paper argues that the application of current reinforcement learning (RL) methods to real-world problems is hampered by the assumption that the actor can be executed at a high frequency (matching that of the actuators). Depending on the architecture of the policy, this can lead to a high computational burden. The authors take inspiration from action chunking observed in the brain to propose an RL algorithm that directly predicts action sequences (or macro actions). After training, it can thus run at slower frequencies than the environment MDP.  Experiments on several simulated continuous control benchmarks demonstrate good performance when predicting macro actions at frequencies lower than that of the MDP, as well as for variable stochastic time steps. The paper furthermore discusses connections of macro actions to neuroscience.

**Strengths:**

The paper relates problems that robotics researchers encounter in a creative way with insights from neuroscience. The issue of high computational demands of modern RL and imitation learning algorithms is quite relevant for the robotics community. The writing is furthermore clear and easy to understand. The proposed algorithm is simple, and easy to understand and implement.

**Weaknesses:**

The paper shifts its motivation from real-time control for robotics to biological plausibility over the course of the paper. The requirements of robotics and biological plausibility are not aligned well, however. Running policies at high frequencies is not necessarily an issue in robotics as long as the computational resources are sufficient. Furthermore, RL algorithms like Soft Actor Critic (SAC) usually do not aim at being biologically plausible, and the use of backpropagation through time in SRL could be viewed as not biologically plausible.

In particular, the small MLP policies trained with SAC should run at about 1 ms on modern hardware (or thereabouts). What is more, even when running them with an action repeat corresponding to a policy execution frequency of 50 Hz, the resulting policies usually perform fine [1]. Hence, the computational load of the experimental set up does not pose a challenge to modern hardware and is therefore not well aligned with the motivation.

What is furthermore absent from the discussion of the results is actual inference times. According to the abstract, reducing the time it takes to produce an action is a main motivation for the paper. However, it is entirely unclear why computing the first action of the macro action should be faster with the proposed algorithm than with vanilla SAC. Hence, it is unclear which advantage the algorithm in its current form offers in a real-world scenario where primitive actions have to be supplied at a fixed frequency to the actuators. It might be possible to run at a higher frequency if a delay is introduced to account for the time needed to calculate the first action of a macro action. Discussing this trade off would be interesting.

The motivation of the paper moreover hinges on the claim that existing methods cannot deal well with low control frequencies in the presence of multiple action dimensions (line 197). In my opinion, this claim is unsubstantiated for two reasons: (i) There is a complete lack of baselines targeted at low control frequencies in the experiments. At the very least showing results for SAC with action repeat is an absolute necessity to put the results into perspective, in my opinion. (ii) The two prior works mentioned (FiGAR and TempoRL) do have experiments on continuous control environments with multiple action dimensions in which they do achieve competitive performance. They should also be included as baselines (TempoRL makes an appearance in the appendix and performs well but is not shown in the main text). The evaluation of SAC is furthermore entirely unfair as it was trained without any action repetition but is then evaluated at a substantially lower frequency. This cannot work and is not a relevant baseline (unlike SAC with action repeat, training with randomized time steps, FiGAR, and TempoRL). For these reasons, I think the experimental results lack relevant baselines and cannot be evaluated properly without them. I would suggest adding these baselines (in the main text).

The field of hierarchical RL is only mentioned in passing in the paper but offers many relevant approaches to easing the computational requirements for control. A higher level (or manager) modulates the lower level (or worker) which produces primitive actions. As the worker may have a simpler architecture, it can run at a higher frequency, while the manager runs at a lower frequency. The worker can be both open loop or closed loop. In practice, simple PD controllers are often used in conjunction with low-frequency policies in robotics and often provide good results. Discussing the hierarchical RL literature to some extent therefore seems necessary.

In line 055 the claim “When RL agents are constrained to human-like reaction times, even state-of-the-art algorithms struggle to perform in simple environments.” is made. This needs a citations or experiments to back this up.

[1] Allshire, Arthur, et al. "Transferring dexterous manipulation from gpu simulation to a remote real-world trifinger." 2022 IEEE/RSJ International Conference on Intelligent Robots and Systems (IROS). IEEE, 2022.

**Questions:**

* In line 098 the claim “Our model learns open-loop control utilizing a slow hardware and low attention, and hence also low energy.” is made. Why should that be true?
* In line 150 the claim that online planning is not feasible in robotics or the brain is made. In my experience this is not true. How did the authors come to this conclusion?
* What is meant by “reducing actor sample complexity” in line 028 in the abstract? Sample complexity usually refers to the amount of transitions needed for training.
* In the paragraph starting in line 050, the authors compare the performance of RL agents on MuJoCo tasks with the performance of humans on complex tasks and relate to reaction times. Why would these settings be comparable as the embodiment, action representation, and tasks are completely different?
* In table one the < sign in the last row should be >.
* What is meant by online planning in the paper? And how is it separate from MPC? It is not clear to me what online planning means.
* In line 054 there seems to be a typo, a redundant “the shows”.
* In line 169 the statement that MPC is limited to systems that have already been modeled accurately is made. However, algorithms like TD-MPC2 use MPC successfully with (partially) learned models.
* The next paragraph states that MPC requires very short timesteps to work. However, that many robotics papers use high frequencies does not mean that it is strictly required. For example, [1] does MPC over skills which has a very low frequency.
* The training curves in the appendix are plotted on top of each other such that it is almost impossible to make out some of the algorithms. Would it be possible to improve these plots?
* Would it be possible to visualize the action sequences predicted by SRL compared to those of SAC?
* In line 516 the claim “In deterministic environments, a capable agent should achieve infinite horizon control for tasks like walking and hopping from a single state.” is made. This is infeasible in environments with complex, possibly unstable dynamics since errors do not disappear but can instead compound and blow up. I would recommend revising this paragraph.

[1] Shi, Lucy Xiaoyang, Joseph J. Lim, and Youngwoon Lee. "Skill-based model-based reinforcement learning." arXiv preprint arXiv:2207.07560 (2022).

---

> ### Author Response · Authors · 2024-11-22
> **Response to Reviewer pfCy**
>
> ## Weaknesses
>
> ### Running Policies at High Frequencies in Robotics
> While high-frequency hardware may alleviate some challenges, our work addresses scenarios where computational resources are insufficient. These scenarios include low-compute hardware, costly observation processing (e.g., images or LIDAR), and low-bandwidth communication. The existence of high-frequency hardware does not negate the utility of our method, which performs comparably at high frequencies while enabling significant improvements for low-frequency scenarios.
>
> SRL’s ability to function at both high and low frequencies without retraining is a key contribution. As demonstrated, SRL maintains performance at high frequencies and provides significant advantages under low-frequency constraints.
> ***
> ### Biological Plausibility
> Our work focuses on system- and algorithm-level biological plausibility, inspired by action chunking observed in the brain. While mechanistic plausibility (e.g., avoiding backpropagation through time) is a separate area of research, it is beyond the scope of this paper. We have also further removed references to biological plausibility to address this issue
> ***
> ### Inference Time Discussion
> We clarify that reducing inference time is not the primary motivation of our paper. Instead, we aim to address control challenges at low decision frequencies. The inference time of the first action in a macro action is comparable to SAC. However, if subsequent observations are unavailable (due to low sensor frequency or costly observation processing), SAC fails, as demonstrated in our experiments.
>
> If you found text in our abstract or elsewhere that suggested a focus on inference time reduction, please let us know so we can correct it.
>
> ***
> ### Claim Regarding Low Control Frequencies
> We have added SAC-J (action repeat) as a baseline, and SRL significantly outperforms SAC-J across all environments except Swimmer, where repetitive actions are less detrimental. Additionally, we address the results reported in FiGAR and TempoRL:
>
> FiGAR: Demonstrates lower scores (947.06 on Ant and 3038.63 on Hopper) compared to SRL’s 6500 and 3500 on these tasks, indicating non-competitive performance.
>
> TempoRL: Limited to Pendulum-v0
>
> Furthermore, these results cannot be extended to variable action sequence lengths required in FAS settings. Results from our implementation (based on TD3) (Figure 8) show TempoRL’s average action sequence length is 2.6, far below SRL’s ability to optimize for longer sequences.
> We hope this clarifies the claim that existing methods are not well-suited for low control frequencies in FAS settings.
>
> ***
> ### Discussion of Hierarchical RL
> We acknowledge that hierarchical RL offers relevant approaches, such as manager-worker architectures. While SRL could be integrated with such methods, our work focuses specifically on generating temporally consistent action sequences to address low-frequency control. Due to space constraints, we limit our discussion to directly related approaches.
> ***
> ### Specific issues in text
> Line 055: We have supported this claim with experiments (SAC-J) and additional reference.
>
> ****
> ## Questions:
> > In line 098 the claim “Our model learns open-loop control utilizing a slow hardware and low attention, and hence also low energy.” is made. Why should that be true?
>
> Energy efficiency relates to reduced attention and slower hardware requirements, both commonly associated with low-energy systems.
>
> > In line 150 the claim that online planning is not feasible in robotics or the brain is made. In my experience this is not true. How did the authors come to this conclusion?
>
> In Appendix Section A.7, we explore the neural basis for sequence learning in humans. Specifically, [2] demonstrates that after training, executing the same action sequences requires fewer neural activations. This reduction in activation suggests that the brain may employ a mechanism akin to SRL, commonly referred to as action chunking. If a model were used for sequence generation after training, the activation levels would remain consistent, which is not observed.
>
> Furthermore, biological neurons typically operate at firing rates of 10–50 Hz. Considering the multiple neuronal layers required to process observations, query a model, and produce actions, it becomes implausible for the brain to rely on high-frequency model queries to achieve temporally precise action sequences. Instead, it likely employs precomputed sequences or learned open-loop strategies.
>
> This principle can be extended to robotics. Current robotics hardware addresses these challenges by leveraging GPUs, high wall power, or large-capacity batteries to maintain high-frequency control. However, adopting principles from SRL could pave the way for robots that are less dependent on costly, fast, and energy-intensive hardware. By enabling efficient control with low-frequency decision-making, SRL offers a potential path toward more sustainable and versatile robotic systems.

---

> > ### Author Response · Authors · 2024-11-22
> >
> > > What is meant by “reducing actor sample complexity” in line 028 in the abstract? Sample complexity usually refers to the amount of transitions needed for training.
> >
> > This refers to the number of observation samples required during control, akin to sample complexity in training. SRL reduces actor sample complexity by producing multiple actions per observation.
> >
> > > In the paragraph starting in line 050, the authors compare the performance of RL agents on MuJoCo tasks with the performance of humans on complex tasks and relate to reaction times. Why would these settings be comparable as the embodiment, action representation, and tasks are completely different?
> >
> > We agree. MuJoCo tasks are much simpler and benefit from noiseless, ground-truth observations, which makes direct comparisons with human tasks challenging. However, the stark disparity—where human reaction times are orders of magnitude slower yet still enable performance in more complex tasks—highlights an intriguing contrast.
> >
> > While we acknowledge the inherent differences in embodiment, action representation, and task complexity, this surprising reversal of expectations warrants exploration. Intuitively, more complex settings would demand faster reaction times and higher frequencies, making the observed discrepancy all the more compelling and worth comparing.
> >
> > > What is meant by online planning in the paper? And how is it separate from MPC? It is not clear to me what online planning means.
> >
> > Online planning refers to generating a sequence of actions using a model rather than relying on ground truth states. This approach enables the agent to execute the planned sequence in the environment without needing to sample directly from it at each step.
> >
> > Model Predictive Control (MPC) is a specialized form of online planning. Unlike general online planning, MPC optimizes the sequence of actions to achieve a specific outcome, goal, or constraint. However, MPC only executes the first action in the sequence before replanning, repeating the optimization process at each step. This iterative optimization introduces additional computational demands not required in simpler model-based online planning approaches.
> >
> > > In line 054 there seems to be a typo, a redundant “the shows”.
> >
> > Thank you for pointing it out. It has been fixed.
> >
> > > In line 169 the statement that MPC is limited to systems that have already been modeled accurately is made. However, algorithms like TD-MPC2 use MPC successfully with (partially) learned models.
> >
> > We removed the line to address this.
> >
> > > The next paragraph states that MPC requires very short timesteps to work. However, that many robotics papers use high frequencies does not mean that it is strictly required. For example, [1] does MPC over skills which has a very low frequency.
> >
> > We could not find any specific mention of the control frequency in the referenced material. Could you kindly point us to the relevant section or provide additional details? This would help us ensure a more accurate and informed response. Thank you!
> >
> > > The training curves in the appendix are plotted on top of each other such that it is almost impossible to make out some of the algorithms. Would it be possible to improve these plots?
> >
> > We have improved visibility by reducing line thickness.
> >
> > > Would it be possible to visualize the action sequences predicted by SRL compared to those of SAC?
> > We apologize for not explicitly mentioning this in the paper. The supplementary materials already contain videos showcasing SRL and SAC on the Lunar Lander environment.
> >
> > Due to file size restrictions, we could not include videos for all environments in the supplementary materials. However, we plan to include these additional visualizations in the final version of the paper to provide a more comprehensive comparison across various tasks.
> >
> > > In line 516 the claim “In deterministic environments, a capable agent should achieve infinite horizon control for tasks like walking and hopping from a single state.” is made. This is infeasible in environments with complex, possibly unstable dynamics since errors do not disappear but can instead compound and blow up. I would recommend revising this paragraph.
> >
> > We have changed the line to In noiseless deterministic environments
> > ***

---

> > > ### Author Response · Authors · 2024-11-22
> > >
> > > While much of your review emphasizes robotics, SRL’s applications extend beyond this domain. Examples include:
> > >
> > > *Low Compute Devices:* Edge devices with constrained computational power often face delays in processing observations (e.g., images or videos). SRL can output multiple actions per observation, compensating for slow processing speeds.
> > >
> > > *Prohibitive Observation Costs:* Some applications involve costly or infrequent observations, such as health monitoring where periodic interventions are based on intermittent blood tests. SRL can effectively operate in such settings. This setting was originally proposed in [3].
> > >
> > > *Remote Control with Low Bandwidth:* Low-bandwidth communication between sensors and controllers creates challenges for real-time control. SRL can adapt to these conditions by generating sequences of actions between observations. See Challenge 7 on [4]
> > >
> > > *Human-Like Control:* By constraining the decision frequencies, we can train agents to learn policies that have a more human-like profile. This is especially of importance to the HCI and gaming community [5].
> > >
> > > ***
> > >
> > > We sincerely appreciate your thoughtful assessment and valuable suggestions, which have greatly helped us refine and improve the manuscript. We have worked diligently to address the concerns you raised and believe the revisions present a compelling case for reconsideration of the manuscript’s rating. Thank you once again for your time, effort, and insightful feedback—it has been instrumental in strengthening our work.
> > >
> > > References:
> > >
> > > [2] Nicholas F Wymbs and Scott T Grafton. The human motor system supports sequence-specific representations over multiple training-dependent timescales. Cerebral cortex, 25(11):4213–4225, 2015.
> > >
> > > [3] Hansen, Eric, Andrew Barto, and Shlomo Zilberstein. "Reinforcement learning for mixed open-loop and closed-loop control." Advances in Neural Information Processing Systems 9 (1996).
> > >
> > >
> > > [4] G. Dulac-Arnold, N, Levine, D. J. Mankowitz, J. Li, C. Paduraru, S. Gowal, and T. Hester. "An empirical investigation of the challenges of real-world reinforcement learning." arXiv, 2020.
> > >
> > > [5] Milani, Stephanie, et al. "Navigates like me: Understanding how people evaluate human-like AI in video games." Proceedings of the 2023 CHI Conference on Human Factors in Computing Systems. 2023.

---

> ### Comment · Reviewer_pfCy · 2024-11-25
> **Response to rebuttal**
>
> ### Inference Time Discussion
> > We clarify that reducing inference time is not the primary motivation of our paper. Instead, we aim to address control challenges at low decision frequencies. The inference time of the first action in a macro action is comparable to SAC. However, if subsequent observations are unavailable (due to low sensor frequency or costly observation processing), SAC fails, as demonstrated in our experiments.
>
> > If you found text in our abstract or elsewhere that suggested a focus on inference time reduction, please let us know so we can correct it.
>
> A major part of the motivation of the paper is the argument that high decision frequencies and low reaction times are impractical in real-world settings (e.g. second sentences of the abstract, line 47 ff or line 75 ff in the introduction) due to high hardware requirements. SRL is presented as a solution to this problem. The issues here is that to supply the actuators with input at the required high frequency, SRL would have to produce the first action at an inference time smaller than the inverse of the actuator frequency. Failing to do so would introduce a delay that is not accounted for in the experiments and that would require additional care. In this sense, inference times and actuator frequencies are related. As the first action requires an inference time comparable to SAC, SRL has the same hardware requirements as SAC, contrary to the motivation of the paper.
>
> In my opinion, this is a big issue that is still not addressed in the paper.
>
> Dealing with missing observations during inference is a more suitable motivation for SRL but it is not very apparent in the abstract and introduction. This line of motivation would also require properly discussing related work that deals with missing observations which is not the case so far.
>
> ### Claim Regarding Low Control Frequencies
>
> > We have added SAC-J (action repeat) as a baseline …
>
> Thank you, I appreciate the additional baseline! However, I do not think FAS is a fair metric in this case as SAC-J is trained with a fixed action repeat but is evaluated with different action repeats. When SAC-J is evaluated with the same action repeat, then it seems to be quite competitive in some cases, for example on InvertedDoublePendulum-v2, or even better than SRL on InvertedPendulum-v2 (see Figure 2). If FAS is to be used as an evaluation metric, then training with variable action repeats (as suggested before) would make more sense, in my opinion.
>
> In the plot corresponding to Pendulum-v1 in Figure 2, SAC-4 is compared to SRL-16. This seems particularly misleading because SAC-4 cannot work well for larger action repeats it never saw during training.
>
> > FiGAR: Demonstrates lower scores (947.06 on Ant and 3038.63 on Hopper) compared to SRL’s 6500 and 3500 on these tasks, indicating non-competitive performance. TempoRL: Limited to Pendulum-v0
>
> FiGAR: The mentioned values are competitive with the base RL algorithm FiGAR was built on (TRPO). It is not clear how FiGAR would do on these tasks when a modern off-policy RL algorithm would be used instead. FiGAR and TempoRL are the most relevant baselines so it would have been interesting to run them on the considered tasks.
>
> ### Specific issues in text
>
> > Line 055: We have supported this claim with experiments (SAC-J) and additional reference.
>
> The reference to Figure 5 in Dulac-Arnold et al. 2020 does not fit the context of the sentence. The sentence is on reaction times but Figure 5 in the cited arxiv version relates to the observation dimension. The experiments on SAC-J also do not directly relate to reaction times as there is no delay between receiving an observation and outputting the first action.

---

> ### Comment · Reviewer_pfCy · 2024-11-25
> **Response to rebuttal part 2**
>
> ### Questions:
>
> > In line 098 the claim “Our model learns open-loop control utilizing a slow hardware and low attention, and hence also low energy.” is made. Why should that be true?
>
> This is related to the Inference Time Discussion. If the inference time is the same as SAC, then the hardware requirement is the same.
>
> > In line 150 the claim that online planning is not feasible in robotics or the brain is made. In my experience this is not true. How did the authors come to this conclusion?
>
> I am still not convinced that this claim holds in this generality. Humans clearly do plan, and so do some approaches used in robotics. Whether this is feasible depends on a lot of factors, for example how reactive the agent has to be or how energy efficient but in general planning is feasible.
>
> >  > The next paragraph states that MPC requires very short timesteps to work. However, that many robotics papers use high frequencies does not mean that it is strictly required. For example, [1] does MPC over skills which has a very low frequency.
>
> > We could not find any specific mention of the control frequency in the referenced material. Could you kindly point us to the relevant section or provide additional details? This would help us ensure a more accurate and informed response. Thank you!
>
> In section 3.4 the authors state “Our skill dynamics model requires only 1/H dynamics predictions and action selections of the flat model-based RL approaches […]”. In table 2 in the appendix the value 10 is given for the hyperparameter 10. So MPC runs at a frequency 10 times lower than the MDP frequency.
>
> > > Would it be possible to visualize the action sequences predicted by SRL compared to those of SAC? We apologize for not explicitly mentioning this in the paper.
>
> > The supplementary materials already contain videos showcasing SRL and SAC on the Lunar Lander environment.
>
> Apologies for not being more precise. I intended to suggest that the action sequences themselves could be visualized, for example in a line plot, and contrasted to those SAC chooses.
>
> > > In line 516 the claim “In deterministic environments, a capable agent should achieve infinite horizon control for tasks like walking and hopping from a single state.” is made. This is infeasible in environments with complex, possibly unstable dynamics since errors do not disappear but can instead compound and blow up. I would recommend revising this paragraph.
>
> > We have changed the line to In noiseless deterministic environments
>
> The point about compounding errors still holds. No learned model is perfect so errors in the state prediction will necessarily occur. Depending on the dynamics of the environment, those errors can blow up, rendering the action sequence ineffective (the agent would trip over after a short time). Dynamics which are not stable are known to be common and lead to these issues. I therefore still think what you are intending to do (walking without sensory input) is not possible. (Also see reviewer thGh’s comment on this.)
>
> > While much of your review emphasizes robotics, SRL’s applications extend beyond this domain. Examples include: […]
>
> I agree that these points are quite relevant for the motivation of SRL (except for the ‘Low Compute Devices’ for the reasons I discussed above). However, they are not used as a motivation in the paper. Instead, a motivation that, in my opinion, does not correspond to what SRL can actually deliver is used (saving compute in real-time control).
>
> My current assessment of the revised paper is:
> * The material present in the paper (method and experiments) is quite interesting and relevant. With the added experiments on SAC-J, this passes the bar for me.
> * The motivation for the algorithm is still somewhat misleading, as SRL does not allow running at higher frequencies compared to SAC without introducing a delay (which is not discussed).
> * There are still claims in the paper that are, in my opinion, not sufficiently substantiated or even incorrect. For example in line 070: “ Consequently, in embodied agents, all components—sensors, compute units, and actuators—are synchronized to operate at the same frequency. Typically, this frequency is limited by the speed of computation in artificial agents.” To the best of my knowledge, this is not true in robotics. It is quite common that different sensors, the actuators and an RL policy run at different frequencies. As an example, consider [1], where the policy runs at 50 Hz, the actuators at 1 kHz, and the object tracking at 10 Hz. I would advise against making such claims without reviewing the robotics literature first.
> * I agree with reviewer YC71 that the overall quality of the paper (positioning with respect to related work, citations for claims, quality of figures) is still somewhat below the threshold for acceptance.

---

> ### Comment · Reviewer_pfCy · 2024-11-25
> **Response to rebuttal part 3**
>
> Overall, the paper feels like it started out motivated by biological observations, changed its motivation to real-time control on low-resource hardware at some point, and then starts to swerve in the direction of dealing with missing observations in the main text (only in this context averaging over the performance of one policy over a wide range of control frequencies makes sense). This structure is suboptimal, in my opinion.
>
> [1] Allshire, Arthur, et al. "Transferring dexterous manipulation from gpu simulation to a remote real-world trifinger." 2022 IEEE/RSJ International Conference on Intelligent Robots and Systems (IROS). IEEE, 2022.

---

> > ### Author Response · Authors · 2024-11-25
> >
> > > A major part of the motivation of the paper is the argument that high decision frequencies and low reaction times are impractical in real-world settings (e.g. second sentences of the abstract, line 47 ff or line 75 ff in the introduction) due to high hardware requirements. SRL is presented as a solution to this problem. The issues here is that to supply the actuators with input at the required high frequency, SRL would have to produce the first action at an inference time smaller than the inverse of the actuator frequency. Failing to do so would introduce a delay that is not accounted for in the experiments and that would require additional care. In this sense, inference times and actuator frequencies are related. As the first action requires an inference time comparable to SAC, SRL has the same hardware requirements as SAC, contrary to the motivation of the paper.
> >
> > > In my opinion, this is a big issue that is still not addressed in the paper.
> >
> > > Dealing with missing observations during inference is a more suitable motivation for SRL but it is not very apparent in the abstract and introduction. This line of motivation would also require properly discussing related work that deals with missing observations which is not the case so far.
> >
> > This issue is already addressed in Section 2. Delay and frequency are distinct challenges, and a complete solution for implementing a control policy on a low-compute robot must address both. As noted in the paper, numerous works focus on delay, any of which can be combined with SRL. However, delay is not the focus of our work. Importantly, existing delay-focused methods do not address low decision frequencies, which is the gap SRL fills.
> >
> > Moreover, SRL can also be implemented in a predictive manner to mitigate delays. For instance, if SRL outputs 16 actions for a single state, the 16th action can be executed while the state for the next set of 16 actions is being processed. Subsequent actions (e.g., the second action) can then follow the 16th action. This allows the action frequency to be higher than the decision frequency while addressing the delay. Properly designed predictive actions can effectively eliminate the need for delay-aware computation.
> >
> > Finally, we want to reiterate that the primary goal of this work is to present a general solution for problems that can benefit from low-decision-frequency approaches, such as observational dropout, low-compute devices, and related challenges. While future work can delve into comprehensive solutions for individual problems, such exploration is beyond the scope of this study. Nonetheless, SRL provides a viable pathway to address low-compute devices, and, as discussed, this would not be achievable with current methods without an SRL-like approach.
> >
> > We hope this addresses your question regarding the motivation of our work. However, if you still disagree with our perspective but recognize the importance of this contribution, we kindly ask if you could suggest specific changes or improvements we can make to the manuscript to achieve a higher rating. Your feedback is invaluable to us, and we greatly appreciate your time and insights.
> >
> > > Thank you, I appreciate the additional baseline! However, I do not think FAS is a fair metric in this case as SAC-J is trained with a fixed action repeat but is evaluated with different action repeats. When SAC-J is evaluated with the same action repeat, then it seems to be quite competitive in some cases, for example on InvertedDoublePendulum-v2, or even better than SRL on InvertedPendulum-v2 (see Figure 2). If FAS is to be used as an evaluation metric, then training with variable action repeats (as suggested before) would make more sense, in my opinion.
> >
> >
> > Training with variable action repeats is not feasible for SAC, as it relies on the assumption that the environment is stationary. Variable action repeats violate this assumption, making it impossible for SAC to learn effectively. This is a well-known limitation. Specifically, it becomes challenging to estimate the value of a state or the Q-value of a state-action pair because the value can vary significantly depending on the duration for which the action is executed.
> >
> > > In the plot corresponding to Pendulum-v1 in Figure 2, SAC-4 is compared to SRL-16. This seems particularly misleading because SAC-4 cannot work well for larger action repeats it never saw during training.
> >
> > SAC-4 is selected because it demonstrates the highest FAS score among the tested configurations. This choice would only be misleading if SAC-16 outperformed SAC-4 or if we failed to report its performance. However, as shown in Figure 5, SAC-16 never achieves a score higher than -1000, making it an unsuitable benchmark for meaningful comparison.

---

> > > ### Author Response · Authors · 2024-11-25
> > >
> > > > FiGAR: The mentioned values are competitive with the base RL algorithm FiGAR was built on (TRPO). It is not clear how FiGAR would do on these tasks when a modern off-policy RL algorithm would be used instead. FiGAR and TempoRL are the most relevant baselines so it would have been interesting to run them on the considered tasks.
> > >
> > > As mentioned, FiGAR and TempoRL cannot be used in the FAS setting because they are not designed to operate under arbitrary decision frequencies. This limitation makes them unsuitable and incomparable as benchmarks in the FAS setting.
> > >
> > > > The reference to Figure 5 in Dulac-Arnold et al. 2020 does not fit the context of the sentence. The sentence is on reaction times but Figure 5 in the cited arxiv version relates to the observation dimension. The experiments on SAC-J also do not directly relate to reaction times as there is no delay between receiving an observation and outputting the first action.
> > >
> > > We apologize for the confusion. The reference was intended to point to Figure 5 in the appendix of our work, and we have updated the text to clarify this. Additionally, Challenge 7 and Figure 13 in the Dulac-Arnold et al. 2020 paper are the appropriate references supporting this point.
> > >
> > >
> > > > This is related to the Inference Time Discussion. If the inference time is the same as SAC, then the hardware requirement is the same.
> > >
> > > As mentioned above, this assumption is incorrect. Inference time and decision frequency are two distinct concepts, both of which impact hardware requirements. Inference time delay can be mitigated using a predictive approach with SRL or by employing a variety of existing solutions applicable to both SAC and SRL. However, SRL uniquely provides a solution for low decision frequency, addressing a critical gap that existing methods do not.
> > >
> > > > I am still not convinced that this claim holds in this generality. Humans clearly do plan, and so do some approaches used in robotics. Whether this is feasible depends on a lot of factors, for example how reactive the agent has to be or how energy efficient but in general planning is feasible.
> > >
> > > Neither are we, that is why we do not make a strong general claim. We talk about **online** planning and not all planning in general. Furthermore, we also mention that it is **non-viable for energy and compute constrained agents**.
> > >
> > > > In section 3.4 the authors state “Our skill dynamics model requires only 1/H dynamics predictions and action selections of the flat model-based RL approaches […]”. In table 2 in the appendix the value 10 is given for the hyperparameter 10. So MPC runs at a frequency 10 times lower than the MDP frequency.
> > >
> > > This still does not demonstrate that it is very low frequency as you claim unless we know the exact control frequency and if it is actually in a similar realm of low frequency (< 20Hz) as presented in this work.
> > >
> > > > Apologies for not being more precise. I intended to suggest that the action sequences themselves could be visualized, for example in a line plot, and contrasted to those SAC chooses.
> > >
> > >
> > >
> > >
> > > We will add it to the final version. However, we are unsure how it will add to the existing results since in actions can vary significantly even in policies trained by the same algorithm with different seeds.
> > >
> > > > We have changed the line to In noiseless deterministic environments. The point about compounding errors still holds. No learned model is perfect so errors in the state prediction will necessarily occur. Depending on the dynamics of the environment, those errors can blow up, rendering the action sequence ineffective (the agent would trip over after a short time). Dynamics which are not stable are known to be common and lead to these issues. I therefore still think what you are intending to do (walking without sensory input) is not possible. (Also see reviewer thGh’s comment on this.)
> > >
> > > We have further changed the text to address this issue:
> > > > In noiseless deterministic environments, a capable agent should achieve infinite horizon control
> > > for tasks like walking and hopping from a single state with minimal error corrections.

---

> > > > ### Author Response · Authors · 2024-11-25
> > > >
> > > > > There are still claims in the paper that are, in my opinion, not sufficiently substantiated or even incorrect. For example in line 070: “ Consequently, in embodied agents, all components—sensors, compute units, and actuators—are synchronized to operate at the same frequency. Typically, this frequency is limited by the speed of computation in artificial agents.” To the best of my knowledge, this is not true in robotics. It is quite common that different sensors, the actuators and an RL policy run at different frequencies. As an example, consider [1], where the policy runs at 50 Hz, the actuators at 1 kHz, and the object tracking at 10 Hz. I would advise against making such claims without reviewing the robotics literature first.
> > > >
> > > > This scenario is quite common in RL game agents referenced in our work. Furthermore, we argue that your example highlights compute as the bottleneck. Specifically, the processing of camera observations into object tracking outputs occurs at a low frequency, which is often treated as an observational bottleneck rather than a computational bottleneck. Additionally, this highlight sour point that while observational, computational and actuator frequencies can be different, the algorithmic choice forces synchronization across the system.
> > > >
> > > > While the actuator frequency is specified as 1 kHz, this is a hardware specification, and the actions themselves are not updated at this rate. There is still a synchronization between the updates provided by the policy and the action frequency. Spline functions or other forms of smoothing may modify the outputted action over time, but this is not equivalent to action frequency. Similarly, to our knowledge, a deterministic policy trained by PPO does not change change its output for the same input. Therefore, it is unclear how the reported frequency discrepancy between observation and policy is handled. We believe that it is also pre-synchronization frequency. At any rate, policy frequency is the algorithmic frequency that is addressed and referenced in our work. For the policy, observation and action frequency is synchronized. Our claim is thus not unsubstantiated.
> > > >
> > > > For instance, in [2], GPM is applied to the CARLA environment, where the agent outputs a target location as the action. The car's locomotion is then managed by a separate function that converts the starting position and target position into a sequence of actions to reach the target. While it may seem that actuation occurs at a faster frequency, actions are only updated when the target position changes.
> > > >
> > > > There are inevitably jargon differences between the fields of RL and robotics, which can lead to misunderstandings. We believe this is the case here.
> > > >
> > > > **Final Thoughts**:
> > > > We believe we have further clarified how our algorithm can be implemented on low-decision-frequency hardware with limited compute, either through delay handling or predictive actions. This reinforces that addressing this setting is a sound motivation for our work.
> > > >
> > > > If your remaining concerns are primarily about how the work is written, we are happy to revise it accordingly. Would you consider revising your rating if we include the additional applications we discussed as part of the motivation, or do you still strongly feel that we need to remove all references to robotics from our work? Your guidance would be greatly appreciated.
> > > >
> > > > [2] Zhang, Haichao, Wei Xu, and Haonan Yu. "Generative planning for temporally coordinated exploration in reinforcement learning." ICLR 22.

---

> ### Comment · Reviewer_pfCy · 2024-11-27
> **Response part 1**
>
> Thank you for your detailed response. It is indeed in the discussion of the method and the presentation of the results where I still see some room for improvement. More concretely, I think the paper could be made more accessible to an audience that is interested in applying SRL on real-world problems. I do not think all references to robotics should be removed. However, I would argue in favor of being careful with general claims given how diverse robotics set-ups and control algorithms are. If the remaining points (addressed below) could be worked in the motivation, the discussion of the algorithm, and the presentation of the results, I would be happy to raise my score.
>
> ### Discussion about frequency, delays, and inference times
>
> I agree with your argument that predicting actions ahead of time to allow for observation processing  or possibly other means of dealing with the delay introduced by the processing of the observation leading to the first action can enable SRL to run with a higher frequency than SAC in practice. For this reason, I think this discussion should be in the paper!
>
> While Section 2 mentions frequency and delay it does so from a more biologically motivated perspective and mostly talks about computation in the brain. For readers that are actually interested in trying out SRL on real hardware, this discussion is not sufficient to point them in the right direction for achieving a high frequency and good performance. I think it might be possible to cut the biological motivation a bit and mention content from your response to me instead. For example, there could be a paragraph on practical considerations somewhere in Section 4.
>
> ### FAS as a metric to compare SAC(-J) and SRL
>
> Indeed, SAC is not intended to be evaluated in a non-stationary environment. If the transition dynamics at test time differ from those experienced during training time, performance can be arbitrarily bad. However, this is exactly what evaluating SAC(-J) with FAS does, as the control frequency is changed after training. For this reason, I do not think FAS is a relevant metric to evaluate SAC(-J) with.
> What would be more interesting, is a direct comparison between SAC-J and SRL-J. SRL has the potential to be better here as well, and the interesting question is: How much better is it in practice?
>
> What I meant by training SAC with variable frequencies is not to train it with one frequency after the other but to vary the frequency stochastically during training. This stochasticity could be considered part of the environment which would therefore be stochastic but not non-stationary. Clearly, this setting is very challenging and SAC would probably not do well. However, it would still be a fairer evaluation than FAS applied to SAC with a fixed J.
>
> ### Reference to back up claim about RL agents struggling with large reaction times
>
> Thank you for the clarification. The Dulac-Arnold reference makes sense to me now and so does Figure 5 in the appendix. I am still wondering a bit if reaction time is the right word here because the RL agent reacts instantly after receiving an observation. Maybe “decision frequency” would be more precise.
>
> ### Feasibility of online planning
>
> The sentence containing “making it [online planning] nonviable for energy and computationally constrained agents like the brain and robots” in line 136 still sounds like online planning was not feasible for brains and robots. However, it is feasible in many situations, just not when the computation budget is very low or short reaction times are required. I would recommend rephrasing this sentence.
>
> ### Visualization of predicted action sequences
>
> I was thinking about a simple environment like an inverted pendulum. SRL should anticipate reaching the target configuration which should be reflected in the action sequence, even though it does not receive a new observation that would inform it about reaching it. Showing an easily interpretable quantitative result like this could strengthen the paper and set SRL apart from action repeat but is not strictly required.
>
> ### Claim about all components of RL systems operating at the same frequency
>
> My issue with this sentence
>
> > Consequently, in embodied agents, all components—sensors, compute units, and actuators—are synchronized to operate at the same frequency.
>
> is that it claims that all components in embodied agents operate at the same frequency which they clearly do not. I think a careful rephrasing could avoid misunderstandings here.
>
> For your interest: In the cited paper, the policy is just fed the last available pose estimate for the object. This is not synchronized with the actions the policy outputs. Other parts of the observation (like joint angles, torques etc.) do change at a higher frequency and therefore the actions the policy outputs also change at 50 Hz.

---

> ### Comment · Reviewer_pfCy · 2024-11-27
> **Response part 2**
>
> ### Small remark
>
> The last two subsection in Section 4 are called “Learning Critic” and “Learning Policy”. I think “Learning the Policy” etc. would sound better and would be consistent with the naming of the subsection “Learning the Model”.

---

> > ### Author Response · Authors · 2024-11-28
> >
> > Firstly, we would like to sincerely thank the reviewer for their time and effort in helping us improve our manuscript. Their thorough engagement during this discussion phase—through detailed feedback, active participation in lengthy discussions, and careful consideration of our responses and edits—has been exemplary. We believe they have set a high standard for ICLR reviews, and we deeply appreciate their constructive contributions.
> >
> > In response to the reviewer’s valuable feedback, we have made several substantial revisions and additions to our manuscript to address the concerns raised:
> >
> > 1. **Biological Perspective and Practical Considerations**
> >
> > - We have reduced the emphasis on the biological perspective in the manuscript and incorporated a discussion about frequency and delay in Section 2.
> > - Additionally, we have added a new section on practical considerations in the appendix due to space constraints while ensuring completeness.
> >
> >
> > 2. **Learning Curve Comparisons**
> >
> > At the request of Reviewer Y33V, we included learning curves for SRL and SAC, where both algorithms are trained and evaluated on the same $J$ to allow for direct comparison.
> > Our results indicate that SRL outperforms SAC in almost every environment, with exceptions being InvertedPendulum, InvertedDoublePendulum, and Swimmer. We hypothesize that:
> > - The pendulum tasks are limited by the precision of the learned model, as balancing tasks require higher precision.
> > -  Swimmer benefits from better exploration afforded by extended actions.
> >
> > We also added randomized frame-skipping results for SAC in three environments (see appendix). While this improves FAS for simpler tasks, it reduces performance for more complex environments like Hopper across all sequence lengths. These findings align with the reviewer’s suggestions and further validate our approach. We will add a further results on all environments in the final version of the paper.
> >
> >
> > 3. We revised “reaction time” to “decision frequency”, as suggested, to improve clarity and precision.
> >
> > 4. **Feasibility of Online Planning**
> >
> > We rephrased the relevant sentence to mitigate any potential misunderstandings and better align with the reviewer's concerns.
> >
> >
> > 5. **Synchronization of RL Components**
> >
> > We clarified that synchronization occurs at the algorithmic level, addressing concern.
> >
> > 6. **Comparison with TempoRL**
> >
> > We added results for TempoRL in the appendix to provide further comparisons with existing methods. Although TempoRL cannot be evaluated under the FAS framework, we included average action sequence length and overall performance for reference.
> >
> > These revisions have substantially strengthened our manuscript, and we believe they address the reviewer’s comments comprehensively. Once again, we extend our gratitude to the reviewer for their insightful suggestions and constructive dialogue throughout this process. Their input has been invaluable in improving the quality and clarity of our work.

---

> ### Comment · Reviewer_pfCy · 2024-11-28
> **My concerns are addressed and I have raised my score**
>
> Thank you very much for the revised text, the additional experiments, and the new figures. I really appreciate the effort the authors have put into the paper during the rebuttal.
>
> 1. **Biological Perspective and Practical Considerations**
>
> The revised version of Section 2 is excellent, greatly adds to the motivation of SRL, and clarifies its placement with respect to existing efforts to bring RL to low-compute hardware.
>
> The new section on practical considerations in the appendix is also appreciated.
>
> 2. **Learning Curve comparisons**
>
> The direct comparisons between the learning curves of SAC and SRL are quite interesting and confirm the advantages of SRL over SAC with action repeat. I agree with the authors that there are some environments where repeating actions can improve exploration but it is not necessary for SRL to tackle this challenge as it is more focused on providing high-frequency actions where they are actually beneficial.
>
> I appreciate the added experiment on randomized frame skipping with SAC. The experiments indeed show that this strategy has its limitations and does not scale well to complex environments.
>
> 3. Thank you!
>
> 4. I appreciate the rephrasing. The statement is more precise now, in my opinion.
>
> 5. Thank you for the addition.
>
> 6. **Comparison with TempoRL**
>
> Adding  results on TempoRL is a good addition that makes the comparison of SRL to baselines more thorough. I understand that TempoRL is not suitable for the FAS setting but it is still interesting to compare its performance to SRL.
>
> My concerns about the manuscript have been sufficiently addressed, and I find it to be indeed significantly improved. I have therefore raised my score. Thank you for the additional work you put into the submission!
>
> I would add one final remark: I would encourage the authors to reference the new material in the appendix in the main text in some form. It might otherwise be overlooked by the reader.

---

### Official Review · Reviewer_Y33V · 2024-11-02

**Soundness:** 3
**Presentation:** 2
**Contribution:** 2
**Rating:** 6
**Confidence:** 4

**Summary:**

This paper proposes Sequence Reinforcement Learning (SRL) as a method to lower effective control frequency by predicting multi-step open-loop action sequences. The action sequences are optimized on model rollouts at the given low-level frequency, while the policy only needs to be queried at a lower rate during deployment. The policy can then be deployed over the full open-loop horizon, or re-queried at intermediate points, where total reward obtained across action sequence lengths favors longer SRL variations over SAC.

**Strengths:**

The paper investigates an interesting problem, considering learning of temporally consistent open-loop action sequences that require fewer queries to the policy during deployment. While the approach is reliant on the quality of the underlying learned model, this enables learned MPC-like control at one end (deterministic 1-step) while also accommodating stochastic sampling periods by using open-loop actions until the next query is possible. The authors evaluate the method across several environments, and compare across 5 seeds against the strong SAC baseline. The paper is generally well-written and clearly motivated.

**Weaknesses:**

- The selection of baselines and or environments should be expanded, as the paper makes a quantitative argument of performance. Other model-free agents (e.g. D4PG, etc.) as well as model-based agents would help to better put the results into perspective. It could furthermore be interesting to run these experiments on the DeepMind Control suite as well, as Gym and DMC tasks have interesting differences in their underlying dynamics and resulting agent performance.
- It would be very informative to see how SAC performs when learning actions at the same rates as SRL-J, e.g. by applying action repeats of J steps. If “SAC-J” would significantly underperform SRL-J, it would provide a stronger argument for adding complexity via model learning.
- Potential inspiration for additional real-world-inspired environment modifications can be found in [1]
- I partially agree with the statement in line 334 that SRL-J is at a disadvantage. However, the agent still uses access to J=1 information via model and critic learning and so the learning curves do provide an accurate comparison across time steps.
- Instead of a learned model, it would be interesting to see an ablation with the ground-truth dynamics to determine how well SRL-J works w/o this “handicap” (ideal model)
- The sentence starting in line 160 highlights that SRL has a model complexity of zero after training. It should be made more clear that this actually applies to many MBRL algorithms (e.g. references in lines 138-143).
- Line 204: consider providing a brief summary of the insights of Section 6 here, or moving Section 6 forward. Otherwise it’s difficult to follow insights that have not been presented, yet.
- Sentence repetition in abstract lines 17-19
- Minor: line 49 —> missing period; line 265 —> model not subscripted

[1] G. Dulac-Arnold, N, Levine, D. J. Mankowitz, J. Li, C. Paduraru, S. Gowal, and T. Hester. "An empirical investigation of the challenges of real-world reinforcement learning." arXiv, 2020.

**Questions:**

- What variations of SRL are included in “all the policies” in line 374? If e.g., SRL-2 is included how would it provide open-loop actions for 16 steps if 16 is sampled from the uniform distribution in line 376?
- The setting on Figure 2 when considering the left-most action sequence length appears similar to an MPC setting for the SRL-J agents, where performance over model-rollouts is learned but only the first action is executed before predicting a new sequence. Could you elaborate on the performance delta of SRL-8/16 to SAC? Is this due to poor model fit?
- Could you elaborate on what you mean by the following sentence: “The SAC algorithm addresses this by maximizing the entropy of each action in addition to the expected return, allowing our algorithm to automatically lower entropy for deeper actions farther from the observation.” How does this lower the entropy of “deeper” actions?
- Is there a good argument of why Gym-v2 versions of the tasks were chosen?

---

> ### Author Response · Authors · 2024-11-22
> **Response to Reviewer Y33V**
>
> Thank you for your detailed and insightful review. Your feedback highlights several important points, and we have made significant improvements to the manuscript based on your suggestions. Below, we address your concerns, describe changes made, and provide clarifications where necessary.
>
> ***
> ## Weaknesses
> ### Expanded Baselines and SAC-J
> We have added SAC-$J$ as a baseline in the revised manuscript. Our results show that SAC-$J$ significantly underperforms SRL across all environments, except for the Swimmer environment, where action repetition offers a significant advantage due to better exploration. This comparison strengthens the argument for SRL’s efficacy in the FAS setting, particularly when adapting to low decision frequencies.
>
> ### Generative Planning Method (GPM)
> We have also included GPM as a benchmark and observed that SRL significantly outperforms it across tested environments. While additional model-based and model-free methods could provide further perspective, their inclusion is computationally prohibitive. SAC models alone required approximately 40 GPU-days for training in the presented experiments, and expanding the benchmark suite would result in months of computational time. Not to mention that FAS calculation requires 800 evaluation episodes per environment. We believe this acts as an implicit gatekeeping mechanism and should not detract from the significant improvements demonstrated by SRL. Additionally, most existing methods lack mechanisms to adapt to observational dropout or low decision frequencies, making their relevance to the FAS setting limited.
>
>
>
>
> ### Ablation with Ground-Truth Dynamics
> We agree that an ablation with ground-truth dynamics could provide valuable insights. However, this would require reimplementation of algorithms with parallelized environment simulations to support batched updates. We leave it to future work.
>
>
> ### Other points:
>
>  Thank you for suggesting the Dulac-Arnold et al. paper ([1]). We have added it to our discussion, and it partly addresses concerns about baseline selection. Notably, Challenge 7 in the paper demonstrates using preliminary results that D4PG, like SAC, suffers significant performance degradation at longer timesteps, further supporting the value of SRL for adaptive frequency control.
>
>
> The statement about "model complexity of zero after training" has been revised for clarity. This refers specifically to model-based online planning methods, where a learned model is required at deployment. SRL’s ability to generate action sequences without a model after training distinguishes it from these methods. Model-based data augmentation methods, while having a model complexity of zero after training, do not provide mechanisms for generating sequences of actions based on a single state.
>
> ### Other Revisions
> Lines 204, 49, and 265 have been updated for clarity.
> The repetition in the abstract has been removed.

---

> > ### Author Response · Authors · 2024-11-22
> >
> > ***
> > ## Questions
> > Q1: Variations of SRL in “All Policies” (Line 374)
> > All SRL variations ($J$ =2,4,8,16) are included. SRL-2 can generate open-loop actions for up to 16 steps by leveraging the GRU in the actor network, which allows arbitrary-length action sequences to be computed from a single state input.
> >
> > Q2: Performance Delta Between SRL-$J$ and SAC
> > You are correct that this setting is similar to MPC. This is an excellent question, and we’d like to share our insights into the performance delta between SRL-8/16 and SAC:
> >
> > The short answer is yes, the delta is primarily due to poor model fit. The long answer: SRL does not use model rollouts to compute the n-th step value for training the value function. Instead, the critic in SRL, like SAC, is trained using one-step transitions, meaning model accuracy does not directly impact the critic’s accuracy. However, the actor’s primitive actions are trained using intermediate states predicted via model rollouts, making the subsequent actions in a sequence dependent on the model’s accuracy.
> >
> > The first primitive action in the sequence is largely unaffected by model accuracy, as it doesn’t rely on rollouts. However, inaccuracies in the predicted intermediate states for subsequent actions can propagate back through the GRU, affecting the gradient of the first action as well. This issue is compounded because all primitive actions in the GRU are currently weighted equally during training. Normally, if the model and critic are well-aligned—i.e., the predicted state value is close to the true value predicted by the critic—this alignment facilitates effective training of the GRU. In such cases, optimizing the first action naturally leads to higher-value states being predicted by the model, which in turn improves subsequent actions.
> >
> > When the model and critic are not aligned due to poor model accuracy, particularly for longer sequences, the misalignment negatively affects the entire action sequence, including the first action. Despite this, SRL shows comparative robustness to model accuracy, hypothetically because achieving model-critic alignment is easier than achieving high overall model accuracy. This alignment enables SRL to outperform model-based online planning in environments with large state and action spaces.
> >
> > We also experimented with passing gradients through the model rollouts to train the actor directly, but this approach resulted in worse performance, as it required stricter model-critic alignment. We believe this issue could be mitigated by introducing weighted updates that prioritize the initial actions in a sequence or by further improving model accuracy. These refinements are promising directions for future work.
> >
> > Q3: Entropy of Deeper Actions
> > The statement regarding entropy in deeper actions was inaccurate and has been corrected in the revised manuscript.
> >
> > Q4: Why Gym-v2 Tasks?
> > Gym-v2 tasks were chosen as they matched our version setup. The differences between v2, v3, and v4 tasks do not affect the results, making results comparable across versions.
> >
> > ***
> > Thank you for your thoughtful comments and suggestions. We believe the added benchmarks, corrected inaccuracies, and clarifications meaningfully address the concerns raised. We look forward to your feedback and hope that these revisions improve the overall perception of our work.
> > Are there any remaining concerns preventing a higher recommendation for the manuscript?

---

> > > ### Comment · Reviewer_Y33V · 2024-11-23
> > >
> > > Thank you very much for providing the additional SAC-J experiments to better understand differences to SRL-J. The learning curves in Figure 3 and 4 are difficult to compare. Could you do a 2 row by 6 column grid per “J” (the 12th field can have the legend), and compare SAC to  SRL for a particular “J” (one 2x6 grid for J=[1, 2, 4, 8, 16])? This would allow for much easier comparison.
> > >
> > > The mention of “gatekeeping” is an interesting and important consideration. It probably often depends on what authors want to show and what timeline was chosen to run which experiments to support the claims.
> > >
> > > There can be subtle differences between versions, which is why I was curious (e.g. https://github.com/openai/gym/pull/2595#issuecomment-1099309407).

---

> > > > ### Author Response · Authors · 2024-11-24
> > > >
> > > > Thank you for your prompt response and thoughtful feedback!
> > > >
> > > > We have updated the manuscript to include additional figures showing the learning curves for SRL across different J values (J=[2,4,8,16]). These have been added at the end of the appendix. Note that we do not include J=1, as SRL in this case is equivalent to SAC.
> > > >
> > > > It would indeed be interesting in future work to compare SRL with a faster action frequency than SAC at J=1 by running the SRL training algorithm at a finer timescale than the default. In our current work, we assume that the default timestep is optimized for critic convergence. However, this assumption might not always hold true, and SRL could potentially outperform SAC in the default setting by enabling a higher action frequency while maintaining the same decision frequency. This exploration is left for future work.
> > > >
> > > > Please feel free to let us know if you have any further questions or suggestions. We sincerely appreciate your time and input!

---

> > > > > ### Comment · Reviewer_Y33V · 2024-11-29
> > > > >
> > > > > Dear authors, thank you for providing the additional plots. I believe these are really helpful in better showcasing potential advantages over SAC. General comparison against additional baselines would significantly strengthen the paper, but I'm happy to increase my rating.

---

### Official Review · Reviewer_thGh · 2024-11-03

**Soundness:** 3
**Presentation:** 3
**Contribution:** 3
**Rating:** 8
**Confidence:** 2

**Summary:**

This model introduces Sequence Reinforcement Learning (SRL), a framework inspired by biological learning agents that allows prediction of action sequences of variable length. The method builds on the Soft-Actor-Critic algorithm, comprising a policy and a state-action-value function. Additionally, a model of the environment is trained and used for updating the policy. The policy includes a GRU RNN allowing the computation of arbitrary length action sequences starting from a single observation. While the model and the critic are trained on truly experienced trajectories only, training of the policy is done by predicting intermediate states using the model and the critic for assigning state-action-values. The method is evaluated on several continuous control tasks. The authors introduce a metric they call Frequence-Averaged Score (FAS) which is defined as the area under the curve of the reward vs decision frequency plot, and find that the SRL framework achieves significantly higher FAS-scores compared to the SAC baseline. They show that this metric is useful in predicting the average reward achieved on a randomized timestep version of the environment, arguing that a policy trained with SRL is better equipped for bridging the sim-to-real gap. The paper ends with a comprehensive comparison to structures found in the mammalian brain.

**Strengths:**

The introduced SRL method is explained well and is easy to understand while also showing significant improvement in the introduced FAS score.  Experimental evaluation is good and the usefulness of the FAS score is adequately demonstrated. The proposed SRL framework is well motivated using recent neuroscientific discoveries.

**Weaknesses:**

Demonstrating the improvement in sim-to-real transfer would add to the quality of the paper.
Likewise, including methods that use action repetition and macro-actions could be an interesting addition.

Minor mistakes:
- Line 104: demonstrate**s**
- Line 376: performance of the policy **in** when the frequency is not constant
- Line 535: we introduce**s** the Frequency-Averaged-Score (FAS) metric

Finally, I can't say that I fully agree with the statement "simple tasks like walking can be performed without input states if learned properly". Biological agents also use a range of inputs to maintain gait.

**Questions:**

1. Like SAC, SRL is using batched experience replay. Can you elaborate on how this interferes with biological plausibility of the method?
2. What is the $\alpha$ that shows up in eq. 3? What does the $\alpha \log \pi$ term represent?

---

> ### Author Response · Authors · 2024-11-22
>
> Thank you for your thoughtful and constructive review. We greatly appreciate your positive feedback on the clarity, motivation, and experimental rigor of our work, as well as your suggestions for improvement. Below, we address the specific weaknesses and questions you raised, along with the changes made to the manuscript.
> ***
> ## Weaknesses
>
> ### Sim-to-Real Transfer
> While bridging the sim-to-real gap is an important motivation for SRL, its applications extend beyond this scope. As highlighted in the paper, the SRL framework addresses open problems in reinforcement learning, including observational dropout and adaptive control under varying decision frequencies. Below are additional examples of practical use cases where SRL can be advantageous compared to traditional RL methods:
>
> 1. *Low Compute Devices:* Edge devices with constrained computational power often face delays in processing observations (e.g., images or videos). SRL can output multiple actions per observation, compensating for slow processing speeds.
>
> 2. *Prohibitive Observation Costs:* Some applications involve costly or infrequent observations, such as health monitoring where periodic interventions are based on intermittent blood tests. SRL can effectively operate in such settings. This setting was originally proposed in [1].
>
> 3. *Remote Control with Low Bandwidth:* Low-bandwidth communication between sensors and controllers creates challenges for real-time control. SRL can adapt to these conditions by generating sequences of actions between observations.
>
> ### Action Repetition and Macro-Actions
> We have incorporated a version of action repetition into SAC (denoted SAC-$j$) and included the results in the revised manuscript. These comparisons demonstrate that SRL consistently outperforms SAC-$J$ across all action repetition lengths, highlighting the effectiveness of SRL.
> ***
> ## Minor Errors
> Thank you for pointing out the minor errors. We have corrected them.
>
>
> > Finally, I can't say that I fully agree with the statement "simple tasks like walking can be performed without input states if learned properly". Biological agents also use a range of inputs to maintain gait.
>
> We have revised the statement to:
> "..simple tasks like walking can be performed without input states in noiseless simulations if learned properly."
> ***
>
> ## Questions
> *Q1: Biological Plausibility of Batched Experience Replay*
>
> In this work, we focus on system- and algorithm-level biological plausibility rather than mechanistic biological plausibility. While batched updates (like backpropagation) are not biologically plausible, they are a pragmatic choice for scaling and training efficiency in reinforcement learning algorithms. These methods allow us to explore biologically inspired structures and principles (e.g., hierarchical control and variable decision frequencies) without being constrained by low-level biological constraints.
>
> *Q2: Explanation of $\alpha$ in Equation 3*
>
> The parameter $\alpha$ is the temperature parameter from the original SAC algorithm. It controls the relative importance of the entropy term in the objective function, balancing exploration and exploitation. We have added a clarification to the manuscript.
> ***
> We appreciate your positive assessment and the thoughtful suggestions provided. These have allowed us to refine the manuscript further. The additional experiments, clarified statements, and corrections aim to address your concerns fully. Thank you again for your time and effort in reviewing our work.
>
> ## Question for the Reviewer
> Are there any other concerns that we can address in the current or future revisions?
>
> ### References
> [1] Hansen, Eric, Andrew Barto, and Shlomo Zilberstein. "Reinforcement learning for mixed open-loop and closed-loop control." Advances in Neural Information Processing Systems 9 (1996).

---

> > ### Comment · Reviewer_thGh · 2024-11-24
> >
> > I thank the reviewersfor answering my questions and for including experiments with SAC and action repetition.
> > This adds to the quality of the paper. I therefore raise my score to recommend acceptance.
> >
> > Best of Luck!

---

> > > ### Author Response · Authors · 2024-11-25
> > >
> > > Thank you for your thoughtful review and for raising your score to recommend acceptance. We greatly appreciate your recognition of our efforts and your invaluable feedback, which helped improve our work.
> > >
> > > Thank you again for your support!

---

> ### Comment · Reviewer_thGh · 2024-11-29
> **Additional perspective**
>
> After reading the other reviews and rebuttals, I would like to add some additional points.
>
> While the proposed method was mainly motivated by low compute latency for actuators, the chosen experiments do not require this on modern hardware as reviewer pfCy has pointed out. Here are ideas for further experiments that would convincingly show the need for slow decision frequencies.
>
> - Control from pixels: Processing image data requires extensive computation. Showing that the number of frames that need processing can be reduced dramatically (e.g. only every 16th for SRL-16) may lead to a more convincing argument.
> - MPC baseline: MPC too requires heavy computation. Showing how SRL compares to MPC would further illustrate the advantages of the method. Similarly, using the same kinematic model for SRL is place of the learned one, as suggested by reviewer Y33V, would make a very interesting experiment.
>
> As a final note, I have to admit to be not very knowledgeable in this area of RL that encompasses "frame-skipping, action repetition, long-horizon exploration, and action correlations". Reviewer YC71 is lamenting a lack of discussion of the relevant literature. I can only agree in this regard and would have enjoyed learning more from a more nuanced section 3.3. Nonetheless, the Related Work section in the current manuscript is extensive as is and covers a broad range of backgrounds.
>
> These were some additional remarks to hopefully help deciding on this borderline paper. Rest assured that I will leave my current rating untouched.

---

> > ### Author Response · Authors · 2024-12-01
> >
> > Thank you for providing this additional perspective. These are indeed valuable and insightful suggestions that would further highlight the strengths of SRL. While we are unable to implement them in the current version of the paper, we are committed to exploring these ideas in future work. Specifically, we aim to include a comparison involving RL with image processing or MPC, and we hope to demonstrate these results in a ICLR workshop paper, if not at the main conference.

---

### Official Review · Reviewer_YC71 · 2024-11-06

**Soundness:** 2
**Presentation:** 2
**Contribution:** 2
**Rating:** 6
**Confidence:** 3

**Summary:**

This paper proposes a new model-based RL method that enables a lower frequency for action decision-making, slower than the frequency of sensing and execution. To accomplish this, the proposed method, Sequence RL (SRL), predicts a sequence of actions at each decision step, allowing for a lower decision frequency while employing a high-frequency dynamic model learner and action-value estimator. This setup mimics biological behavior, where the brain’s computation frequency is lower than that of sensing and actuating processes. Additionally, the paper introduces a new metric, FAS, to assess an RL method’s robustness and adaptability across different frequency settings, demonstrating that SRL achieves a high FAS score.

**Strengths:**

- The research motivation of the paper, specifically the comparison of decision patterns and frequencies between RL and humans, is compelling.
- The paper establishes a strong connection to biological fundamentals, providing relevant examples and insights throughout.
- The main idea—designing an RL algorithm inspired by biological principles, where each component operates at different frequencies—is novel, and the storyline leading up to the experiment section is smooth and easy to follow.

**Weaknesses:**

## Lack of related works
- This paper primarily focuses on its connection to biological insights and motivation but overlooks relevant efforts in the RL literature addressing frame-skipping, action repetition, long-horizon exploration, and action correlations. Several studies, such as [1,2,3,4], have explored similar topics from different perspectives. Although these works are not biologically motivated, their contributions are highly relevant to this paper and should not be ignored.

- The way of using GRUs in sequence action generation is very similar to the work in [4], please consider adding discussion.

## Technical Issues
- From line 252 to line 259, it is unclear that why using max entropy RL techniques, such as SAC, can address the issue of additive noise and automatically lower entropy for deeper actions arther from the observation.

- Inproper declaration from line 292 to 296. There are several works that focus on credit assignment for action sequences, such as [2, 5].

- Does the current work fully address the issues listet in line 93? Namely the sparse rewards, jerky control, high compute cost, and catastrophic failure due to missing inputs. If not, it is better to remove this misleading content.

## Poor Experiment Quality (Critical Issue for this Paper)
- The empirical results only include SAC as a baseline method, which was introduced in 2018. This is clearly insufficient. All the methods mentioned above have demonstrated advantages over SAC by employing different techniques for modeling action sequences or predicting temporally correlated actions. These approaches should theoretically achieve good FAS scores as well. Please consider including some of these methods to enrich the experiments and enhance the technical rigor of the paper.

- Even with SAC as the only baseline, the proposed method, SRL, only outperforms SAC in 6 out of 10 tasks. Despite its advantage in the proposed FAS metric, SRL's performance over SAC is marginal and unconvincing.

- In 5 out of 10 tasks (InvPendulum, Hopper, InvDPendulum, Reacher, and Swimmer), the Online Planning method outperforms SRL, contradicting the description in the caption of the "second Table 3" on page 9.

- Many important results are placed in the appendix, while the main paper is disproportionately occupied by the introduction to biological concepts.

## Excessive Discussion of Biological Concepts
- Section 6 should be either removed entirely or moved to the appendix, as it detracts from the paper’s main focus on learning representation rather than biology.

## Minor issues
- It is better to use different math symbols to distinguish the macro action and per-step action, especially in equation 1. Both sides used $a_{t'}$
- There are two "Table 3" captions in the paper: one on page 8 and another on page 9.
- The layout of figures and text in the paper is of low quality and requires adjustments and proofreading. For example, the title of Section 6 is oddly placed.

## References
[1] Raffin, Antonin, Jens Kober, and Freek Stulp. "Smooth exploration for robotic reinforcement learning." Conference on robot learning. PMLR, 2022.

[2] Li, Ge, et al. "Open the Black Box: Step-based Policy Updates for Temporally-Correlated Episodic Reinforcement Learning." ICLR 24.

[3] Chiappa, Alberto Silvio, et al. "Latent exploration for reinforcement learning." Advances in Neural Information Processing Systems 36 (2024).

[4] Zhang, Haichao, Wei Xu, and Haonan Yu. "Generative planning for temporally coordinated exploration in reinforcement learning." ICLR 22.

[5] Ni, Tianwei, et al. "When do transformers shine in rl? decoupling memory from credit assignment." Advances in Neural Information Processing Systems 36 (2024).

**Questions:**

Please address the issues outlined in the weaknesses section. The effectiveness of the proposed method is significantly limited by the low quality of the experiments, particularly the empirical results. It is essential to include extensive comparisons with related works to more convincingly demonstrate the method’s effectiveness, beyond merely relying on the proposed frequency metric.

---

> ### Author Response · Authors · 2024-11-21
> **Response to Reviewer YC71**
>
> Thank you for your detailed and thorough review. Your comments were invaluable in improving the clarity, rigor, and presentation of our paper. Below, we address your concerns point by point and outline the changes made in the revised manuscript.
> ***
> ## Lack of Related Works
>
> We appreciate the reviewer’s suggestions for related works and have incorporated them into the revised manuscript. Below, we address their relevance and relation to our work:
>
> **Frame-Skipping and Action Repetition:** As discussed in Section 3.4, these methods are indeed relevant. Traditionally frame-skipping and action repetition have been studied as methods for better exploration or explainability. In this work however, we suggest that they might be used in place of macro-actions in the FAS setting to act even when the observation is missing.
>
> **Discussion of Suggested Works:**
>
> [1, 2, 3]: These works focus on exploration techniques (e.g., smooth exploration and temporally correlated exploration). However, they do not address the unique challenges of the Frequency-Averaged Score (FAS) setting, where decision frequency is lower than action frequency. Specifically:
> [1]: Generalized state-dependent exploration (gSDE) improves exploration and smoothness but relies on continuous state inputs. In FAS settings, it will have to default to action repetition. While gSDE could enhance SAC exploration during training, it does not address evaluation under observational dropout or lower decision frequencies.
>
> [2, 3]: Similarly, these methods leverage temporal correlations to improve exploration but cannot adapt to scenarios with missing observations during evaluation or low decision frequency.
> [4]: This work (Generative Planning Method, GPM) is indeed closely related. We thank the reviewer for pointing it out and have now included GPM as a benchmark in our experiments. While GPM is primarily focused on exploration, its similarities to SRL make it a valuable comparison. Key differences include:
>
> **Objective:** GPM is designed to enhance exploration, whereas SRL focuses on precision and adaptability under low decision frequencies.
>
> **Training:** GPM trains Q-values for sequences without addressing the instability caused by deeper actions diverging. SRL resolves this issue through model-based training.
>
> **Evaluation:** GPM's reported results do not evaluate full plan execution under observational dropout.
>
> We have now benchmarked GPM in FAS settings, where it performs poorly compared to SRL and even SAC.
> We hope these clarifications and additional experiments demonstrate our consideration of related works and reinforce the novelty of SRL.
> ***
> ## Technical Issues
> Lines 252–259: We agree this section was unclear. The statement has been corrected. Specifically, SAC automatically tunes the entropy parameter for all actions, making it desirable in our setting as it avoids extensive hyperparameter tuning across different sequence lengths.
>
> Lines 291–296: We clarify that our discussion of credit assignment refers specifically assigning credit to primitive actions in an action sequence performed based on a single state, which differs from the settings addressed in [5]. Thus, [5] does not directly apply to our work.
>
> Line 93: Based on your feedback, we have revised the text to focus solely on catastrophic failure due to missing inputs, which our method explicitly addresses.
> ***

---

> ### Author Response · Authors · 2024-11-21
>
> ## Poor Experiment Quality
> Additional Benchmarks: We have added GPM as a benchmark across four shared environments. As reported, GPM performs poorly in FAS settings, further validating SRL’s effectiveness. Additionally, we included results for SAC trained with different timesteps (SAC-$J$) for a comprehensive comparison.
>
> Performance Clarification: We acknowledge the confusion caused by our presentation of learning curves. In the original Figure 5, SRL and SAC were evaluated under different conditions, making direct comparisons difficult. To address this:
>
>
> We have revised the manuscript to include separate plots for SAC-$j$ (Figure 3) and SRL-$𝐽$ (Figure 4). Similarly, we have separated the plots for action sequence length (Figures 5 and 6).
> We clarified that SRL’s learning curves reflect a disadvantageous evaluation condition (state provided every $J$ steps), while SAC (SAC-1 in the revision) is evaluated under standard conditions.
> For a more equitable comparison, we direct the reviewer to Figure 5 and 6 (in the revision), where SAC and SRL are evaluated under identical settings. In these plots, the left-most markers represent the evaluation under the standard setting (i.e., a state is provided at every timestep). By comparing Figures 3, 4, 5 and 6, it becomes evident that learning curves do not fully capture SRL's performance. For example, while SRL-16 appears to exhibit near-zero performance on InvertedPendulum-v2 in Figure 4, Figure 6 shows that SRL achieves optimal results when tested under the standard setting.
>
> This comparison underscores that SRL does not underperform in any environment when evaluated under standard conditions. However, it is important to emphasize that the primary goal of our work is not to enhance performance in standard settings, as SAC already achieves optimal or near-optimal performance across most environments (except Swimmer). Instead, the focus of our work is to improve performance under the remaining 15 markers, which correspond to settings where decision frequency is reduced, and observational dropout occurs. In these scenarios, our algorithm demonstrates substantial improvements over SAC. Additionally, we have amended the text in the manuscript to provide clearer context and avoid misinterpretations.
>
> We hope this explanation, along with the updated figures and text, adequately addresses concerns regarding the significance and interpretation of the results presented in our work.
> ***
> > In 5 out of 10 tasks (InvPendulum, Hopper, InvDPendulum, Reacher, and Swimmer), the Online Planning method outperforms SRL, contradicting the description in the caption of the "second Table 3" on page 9.
>
> We believe that the reviewer is referring to the following statement:
> “We see that SRL can learn action sequences that perform better than model-based online planning in many environments.”
>
> We have changed it to:
>
> “We see that SRL can learn action sequences and is competitive to model-based online planning.”
> ***
> ## Excessive Discussion of Biological Concepts
> Thank you for this suggestion. The biological discussion has been moved to the appendix to streamline the main paper and maintain focus on the RL contributions.
>
> ***
> ## Minor issues:
> We have revised the notation for macro actions to $m$
> Formatting errors have been corrected.
>
> ***
> ## Answer to Questions:
> We have addressed the primary concerns regarding related works, benchmarks, and experimental clarity. Additionally, we emphasize that the FAS metric measures average performance across frequencies and was not specifically designed to favor our method.
>
> ***
> We appreciate your assessment and the thoughtful suggestions provided. These have allowed us to refine the manuscript further. The additional experiments, clarified statements, and corrections aim to address your concerns fully. Thank you again for your time and effort in reviewing our work.
>
> ### Questions for the Reviewer
> 1. Have the additional experiments and clarifications addressed your concerns? Are there other related works we may have missed that would strengthen the manuscript?
>
> 2. If you are unable to change your rating, could you kindly highlight any remaining issues so we may address them in future revisions?

---

> > ### Comment · Reviewer_YC71 · 2024-11-22
> > **Could you please mark the main updates to the paper using blue color?**
> >
> > Hi,
> >
> > Thank you for your updates to the paper and experiments. As I begin reviewing them, I am finding it challenging to differentiate the updated parts from the original content. Could you please mark the main changes, for example, by using blue color? This would help in identifying the modifications and make the review process more efficient.

---

> > > ### Author Response · Authors · 2024-11-22
> > > **Updates changed to blue color**
> > >
> > > Thank you for the quick response! We have updated the manuscript to reflect the changes in blue. Apart from the marked blue changes, we also made changes to the following figures:
> > > Figure 2
> > > Figure 3
> > > Figure 4
> > > Figure 5
> > > Figure 6
> > >
> > > Finally Section A.7 has been moved from the main text to the appendix.

---

> > > > ### Comment · Reviewer_YC71 · 2024-11-23
> > > > **Reply to rebuttal and update score from 3 to 5.**
> > > >
> > > > Thank you for the updated paper and experiments. Compared to the initial version, the quality of the paper has improved significantly. I will increase my score from 3 to 5 to reflect the updates and the efforts made by the authors:
> > > >
> > > > - The proposed method introduces interesting and novel components, particularly the usage of the state evolution model to populate intermediate states for the action sequence.
> > > > - Additional related works have been included, and the GPM method is now further compared in the experimental section.
> > > > - The paper has shifted its focus more towards RL content rather than the biological aspects, which aligns better with the primary audience of ICLR.
> > > > - The existing tables, plots, and text have been polished, which better supports the authors' arguments for the proposed method.
> > > >
> > > > However, I still feel it is insufficient to rate this paper higher. In my view, the overall quality is still marginally below the ICLR standard. Here are my reasons:
> > > >
> > > > - Beyond GPM, there are likely additional methods that could address this research problem, specifically RL with lower decision or sensory frequencies combined with higher action execution frequencies. For instance, I found another relevant work in NeurIPS 2020 [1]. The discussion of related literature in the paper still feels incomplete.
> > > > - While the updated experiments are appreciated, the inclusion of more baseline methods would offer a more comprehensive comparison. That said, given the time constraints of the rebuttal period, I think it would be unfair to request new experiments at this stage.
> > > > - I share some concerns similar to those raised by reviewer pfCy regarding the necessity of deploying a low-responding frequency RL agent (with decision and sensory frequencies akin to humans) in robot-like systems. Additionally, I am not sure whether the chosen gym tasks are the most appropriate for addressing the research question.
> > > > - While the proposed method demonstrates advantages over SAC and GPM, it would be more impactful if it were applied to challenging tasks where SAC and GPM fail entirely using their default settings. The novel aspects of the method, such as the action sequence prediction and the model-predicted state evolution, could potentially shine more in such complex scenarios.
> > > > - (Minor) The quality of the paper's presentation, including the figures and tables, can still be further improved.
> > > >
> > > > I hope these suggestions and critiques are helpful for improving the paper further in future revisions or work.
> > > >
> > > >
> > > >
> > > > ## References
> > > > [1] Bahl, Shikhar, et al. "Neural dynamic policies for end-to-end sensorimotor learning." Advances in Neural Information Processing Systems 33 (2020): 5058-5069.

---

> > > > > ### Author Response · Authors · 2024-11-24
> > > > >
> > > > > Thank you for your prompt and thoughtful response. We deeply appreciate the updated score and your recognition of the efforts we made to refine the paper. Below, we respond to your remaining concerns and provide further context to address your thoughtful feedback.
> > > > >
> > > > > 1. We would like to emphasize that none of the previous works, including GPM and NDP, directly address the research problem we are tackling. NDP, for instance, does not present results for settings with low decision frequencies and does not discuss how its algorithm would behave or adapt in such scenarios. It operates under the assumption that actuator frequency is higher than observation frequency, making it unsuitable for FAS settings, where we test in setting where the observation frequency is lower than default. Additionally, there is no evidence that NDP generalizes across different plan lengths after training, a critical requirement in our framework. For example, it fails to learn at plan-length of 3 and demonstrates significantly lower performance at lengths higher than 5.
> > > > >
> > > > > 2. As we mentioned, none of the previous works explicitly address or even mention the research problem tackled in this paper. This gap is partly due to the prevailing assumption in RL research that a single, constant timestep governs all components of the system. Our work challenges this foundational assumption, and as a result, there are no existing results directly comparable to ours, nor do prior works provide a clear pathway for adapting their algorithms to this setting.
> > > > >
> > > > >     Adapting unrelated methods to our framework is beyond the scope of this paper. Such adaptations would require extensive modifications, hyperparameter search and will likely result in poor performance because these methods inherently assume that the observation frequency remains consistent during training and evaluation. Additionally, evaluating all unrelated approaches to check whether they unintentionally perform well in this setting would require a large-scale comparative-study, which is not within the scope of this work.
> > > > >
> > > > >     Instead, the role of such a contribution is to establish benchmarks and metrics for the community, enabling future research to build upon this foundation. By publishing this work, we aim to open up exploration into this under-studied area, fostering a broader range of approaches to address these challenges.
> > > > >
> > > > >     Unlike research on high-frequency control, which has a wealth of prior work to draw upon for incremental comparisons, work on low-frequency control remains sparse. Our contribution not only fills this gap but also highlights the need for more attention to this critical area in RL and robotics.

---

> ### Author Response · Authors · 2024-11-24
>
> 3. We respectfully disagree with the notion that the necessity of low-frequency RL agents is unproven or unnecessary in robotics. Modern robot hardware has indeed been designed with existing algorithms in mind, which assume fast decision frequencies. This creates a "survivor bias," where only systems that fit these assumptions are published or widely adopted. However, this does not diminish the need for solutions like SRL.
>
>     For instance, [7] highlights challenges similar to those addressed by SRL, including degraded performance at lower frequencies. Additionally, [8], a GitHub repository (albeit 5 years old) for a line-following robot using a Raspberry Pi, provides a concrete example of how limited processing speed hinders RL deployment on affordable hardware. The repository states:
>
>     > The Raspberry Pi 3B+ could only run the PPO algorithm at about 1 FPS. At this frame rate, the donkeycar couldn't get around the track much faster than the OpenCV line-following algorithm.
>
>     Further evidence of failures of RL algorithms on affordable and slow hardware is sparse, as negative results are seldom published.
>
>     While SRL might not completely solve such issues, it offers a substantial improvement over existing methods like PPO by enabling effective control at lower frequencies. For instance, at 1 Hz, SRL achieves a reward of -400 on Pendulum and 3300 on Ant, compared to SAC’s -750 and 1400, respectively. These results demonstrate SRL’s superior performance in low-frequency settings. This provides a strong motivation for further exploration of this setting.
>
>
>     **Addressing the choice of Gym environments:** Our primary focus was to establish benchmarks on environments that are particularly challenging to solve using extended actions. Notably, previous works such as TempoRL, GPM, NDP, and FiGAR have not addressed difficult environments like Humanoid, Ant, Walker, and Half Cheetah. This limitation can be attributed to two main factors:
>
>     a. Unlike simpler robot arm environments, these locomotion tasks demand precise balancing, which requires high-frequency action updates due to their fast dynamics. As a result, these environments do not benefit significantly from action repetition approaches.
>
>     b. Multiple Action Dimensions: Multiple action dimensions are difficult for open-loop action repetition since it requires the repetition to be  synchronized across multiple degrees of freedom which is unsuitable for tasks like locomotion.
>
> 4. Our algorithm addresses scenarios where the optimal timestep is not feasible. However, since it is built on SAC, it requires SAC to succeed for at least one choice of timestep. If there is an environment where SAC fails across all timestep choices, SRL would also fail.
>
>     It is worth noting that the default timestep varies significantly across the environments we tested and is typically chosen based on its optimality for traditional RL setups. For example, the default timestep is 8 ms for Hopper and Walker and 50 ms for Ant and Half Cheetah. SAC exhibits poor performance on Hopper and Walker at 50 ms, whereas SRL demonstrates near-optimal performance.
>
>
> Thank you once again for your thoughtful remarks and the opportunity to improve our work. While we may differ on certain aspects, your comments have been instrumental in refining the manuscript and addressing key gaps in our results and motivation. Your feedback has been invaluable, and we sincerely appreciate the time and effort you have dedicated to reviewing our work. Thank you once again for helping us improve our contribution to this field.
>
> **References**:
>
> [7] G. Dulac-Arnold, N, Levine, D. J. Mankowitz, J. Li, C. Paduraru, S. Gowal, and T. Hester. "An empirical investigation of the challenges of real-world reinforcement learning." arXiv, 2020.
>
> [8] https://github.com/downingbots/RLDonkeycar

---

> > ### Comment · Reviewer_YC71 · 2024-11-29
> > **Reply to the rebuttal**
> >
> > Hi,
> >
> > Thank you for your reply and explanations to my concerns. I also noticed the long engagement between the authors and reviewer pfCy, which made the storyline clearer, especially for the revised part of Section 2. I would nominate reviewer pfCy for the best reviewer award if there is one.
> >
> > Unfortunately, I remain strict about the overall quality of the paper and will maintain my score at 5. However, I have read and compared all three versions of the paper (initial, rebuttal, and the current version), and I feel this paper is on its way to becoming a good one. Below, I list my reviews for the current version and provide several suggestions for the authors in their future works.
> >
> > **Advantages:**
> > - The method proposed in the paper, namely combining action sequence prediction with model-based RL for intermediate state evolution, is interesting and novel.
> > - The experiments revealed the potential of this approach to address common problems in robot learning, such as low decision frequencies and missing observations.
> > - The revised storyline has reduced its reliance on biological techniques, focusing more on RL and its target research problems.
> > - I agree with the authors in most of their claims regarding research challenges with low-quality hardware.
> >
> > **Shortcomings**
> > - The initial version of this paper appeared half-baked, with poor quality in terms of related works discussion, experiments, and formatting, especially the discussion and comparison with similar methods. I still feel the authors could dedicate more effort to literature research and investigate related works leveraging sequence action predictions. Honestly, I feel I have contributed more to this aspect than the authors themselves. Addressing this would, in my opinion, elevate the paper to a 6-score level.
> > - This paper has shifted its focus from a biologically inspired model to one tailored for low-decision-frequency scenarios. Despite the potential for deployment in low-quality hardware and robots, no such task is included in the paper. Incorporating a small hardware task, such as a locomotion experiment commonly found in conferences like ICRA and CoRL, would make the method more convincing and impactful. With such an addition, I would rate this paper as an 8-score work.
> >
> > **Other Minor Suggestions for Future Work**
> > - Be cautious when claiming that robot manipulations are simpler tasks than locomotion. In fact, manipulations are often more challenging for off-policy and model-based methods due to complex dynamics such as contacts.
> > - Consider redesigning Table 1 to make it smaller and wrapped within the text to save space.
> > - Improve the visualization of Figure 1:
> >   - Consider removing the `BG` and `PFC` labels and emphasizing the RL components if the paper is now more RL-driven than biology-driven.
> >   - Utilize the blank spaces in Figure 1 effectively by rearranging the RL components, possibly incorporating content from Figure 9.
> > - Enhance the visual quality of the learning curves by
> >   - Using larger font sizes.
> >   - Adopting vectorized figures, such as those generated with tikzplotlib.
> > - Consider placing some results from the appendix in the main paper to enhance the presentation of the research gap and method. For instance, include one SAC-J figure in the introduction section to emphasize the low-decision-frequency problem prevalent in common RL methods.

---

> > > ### Author Response · Authors · 2024-11-30
> > >
> > > Thank you for acknowledging the changes and improvements we have made to our work. As ICLR has provided an extended discussion period beyond the manuscript revision deadline, we would like to take this opportunity to further elaborate on our views regarding this matter.
> > >
> > > We sincerely hope that this additional discussion will encourage the reviewer to reconsider their reject recommendation. However, regardless of the outcome, we believe it is valuable to document this exchange for posterity and the benefit of the research community.
> > >
> > > > The initial version of this paper appeared half-baked, with poor quality in terms of related works discussion, experiments, and formatting, especially the discussion and comparison with similar methods. I still feel the authors could dedicate more effort to literature research and investigate related works leveraging sequence action predictions. Honestly, I feel I have contributed more to this aspect than the authors themselves. Addressing this would, in my opinion, elevate the paper to a 6-score level.
> > >
> > > As we noted earlier, none of the works mentioned by the reviewer are directly related to our work. Including comparisons with these approaches would neither enhance the confidence nor the quality of our study. As demonstrated with the GPM results, such approaches are not designed for low-decision-frequency settings and perform poorly in the context discussed in this paper.
> > >
> > > Given this, we struggle to understand why these works are considered relevant to our study. Through our GPM analysis, we demonstrate that exploration-based approaches do not provide a solution for the low-decision-frequency problem and are therefore not directly comparable or related to our work.
> > >
> > > This lack of alignment with prior methods is one reason why our original manuscript included biological inspiration, as no existing approaches directly address the challenges outlined in our study. The reviewer has not pointed to any prior work that addresses this specific setting.
> > >
> > > > This paper has shifted its focus from a biologically inspired model to one tailored for low-decision-frequency scenarios. Despite the potential for deployment in low-quality hardware and robots, no such task is included in the paper. Incorporating a small hardware task, such as a locomotion experiment commonly found in conferences like ICRA and CoRL, would make the method more convincing and impactful. With such an addition, I would rate this paper as an 8-score work.
> > >
> > > We would like to clarify that our focus has not shifted to low-decision-frequency scenarios; the title of our paper reflects our intended scope. The inclusion of biological perspectives was primarily meant to introduce a unifying lens for addressing fragmented challenges in real-time control, such as low-decision frequency, delay, observation dropout, inference times, and limited compute. These issues are often treated as separate subfields in AI control research, where solutions addressing one problem may fail—or even exacerbate—others.
> > >
> > > For instance, while incorporating a high-precision model of the environment might mitigate challenges like delay or observation dropout, it often leads to increased inference times, especially on low-compute devices. When viewed in isolation, such a solution might seem viable, but a broader perspective reveals its limitations. Drawing inspiration from biological systems, which inherently address all these challenges simultaneously, provides a compelling framework for developing more general solutions.
> > >
> > > Furthermore, the field of control using RL exists precisely because of its potential for real-world applications. The MuJoCo benchmarks, utilized in our work, were introduced to evaluate this potential. Algorithms including SAC, TD3, and PPO were proposed with real-world control in mind, and their strengths were demonstrated through simulations using these same benchmarks. Similarly, we demonstrated our algorithm on the MuJoCo environments in this work to highlight the strengths of our algorithm, as is standard practice.
> > >
> > > However, we find it inconsistent and unfair that we are being asked for hardware implementation to validate our results, whereas other works relying solely on MuJoCo simulations have been accepted without such requirements. In fact we highlight the gap in the benchmark with respect to the inconsistent timesteps and address it in our work. Our approach and evaluation align with established norms in the field, and we believe our contributions should be assessed on the same grounds.

---

> > > > ### Author Response · Authors · 2024-11-30
> > > >
> > > > > Be cautious when claiming that robot manipulations are simpler tasks than locomotion. In fact, manipulations are often more challenging for off-policy and model-based methods due to complex dynamics such as contacts.
> > > >
> > > > We apologize for any confusion caused by our poor wording. What we intended to convey is that robotic manipulation tasks are generally less sensitive to slow decision frequencies compared to locomotion tasks that require precise balancing. For example, in the Reacher environment, SAC demonstrates significantly greater robustness to longer time steps than it does in environments involving more dynamic tasks, such as balancing or locomotion.
> > > >
> > > > We also thank the reviewer for their constructive suggestions for further refinement of our work and will incorporate these improvements in the final version.
> > > >
> > > > Finally, we respectfully urge the reviewer to evaluate our work in its current form or, at the very least, provide evidence of any related works we may have missed, as we believe we conducted a thorough literature review prior to embarking on this research.  It is very possible that we might have missed relevant works given the sheer amount of publications in recent years and the absence of unified terminology to address this concern. Additionally, we request that the reviewer consider this recent repost by the ICLR account on X when assessing our work: https://x.com/abeirami/status/1861840261875200078

---

> > > > > ### Comment · Reviewer_YC71 · 2024-12-01
> > > > > **I increase my recommendation score to 6**
> > > > >
> > > > > Hi,
> > > > >
> > > > > Thank you for your thorough engagement and detailed explanations. I now increase my score to 6.
> > > > >
> > > > > Below, I provide my rationale for my previous concerns:
> > > > >
> > > > > ### Why do I think other RL methods predicting sequence actions can be related to this work?
> > > > >
> > > > > - The research question in the current paper focuses on "making RL work in low-decision-frequency systems," with potential applicability to low-compute hardware. In my view, RL methods that predict sequences of actions are highly relevant to this research problem. If a model can predict multiple actions given a single state, the decision frequency can naturally be reduced, facilitating deployment in low-profile hardware. Your method also incorporates a predictor for a sequence of actions to address this problem.
> > > > >
> > > > > - Despite this intuition, I conducted additional literature research and carefully reviewed the techniques proposed by NDP and GPM. I now agree with the authors that these methods are not entirely suitable for this research problem. For instance, NDP can predict a sequence of actions similar to SRL but still relies on intermediate state rollouts from the environment for updates. In contrast, SRL evolves these methods by using a model-based state predictor. Based on this differentiation, I decided to increase my recommendation score to 6.
> > > > >
> > > > > ### Why do I think lacking hardware experiments prevents SRL from being rated as an 8-score paper?
> > > > > For the other RL works mentioned by the authors, the main focus is on stable training and sample efficiency, such as:
> > > > > - PPO: A simplified version of TRPO that enforces trust regions for stable policy updates.
> > > > > - TD3:  Incorporates pessimistic estimation with two Q-networks to prevent value overestimation in off-policy RL.
> > > > > - SAC: Uses a soft objective (entropy bonus) to encourage exploration and prevents Gaussian policy from collapse.
> > > > >
> > > > > In my opinion, these research questions are fundamental to RL, so pure simulation tasks are appropriate for their experimental settings. However, for SRL, a significant portion of the discussion revolves around its potential to address challenges in low-compute hardware, which benefits from low-decision-frequency RL methods. Notably, the keyword "hardware" appears 10 times in the paper. Therefore, I believe adding several hardware tasks would make the contribution more concrete, turning potential into validated experimental results.

---

> > > > > > ### Author Response · Authors · 2024-12-04
> > > > > >
> > > > > > Thank you for your thorough literature review and valuable feedback. Based on the insights gained during this discussion period, we are now working on implementing a hardware-based evaluation of SRL. Additionally, we will also design a more realistic simulation environment, based on the hardware implementation, to reflect low-compute scenarios, incorporating elements such as random delays and low-decision frequency to further validate our approach.
> > > > > >
> > > > > > We sincerely appreciate your stellar participation and thoughtful contributions throughout this discussion period.

---

### Author Response · Authors · 2024-11-21
**Changes in the manuscript**

We sincerely thank the reviewers for their constructive feedback, which has greatly helped improve our manuscript. In response, we have made the following significant revisions:

1. Reduced the discussion of biological aspects throughout the paper. The section titled "Neural Basis for Sequence Learning" has been relocated to the appendix.

2. Included results for SAC trained on different decision frequencies/timesteps, denoted as SAC-$J$.

3. Expanded the related works section to include studies on temporally correlated exploration.

4. Made minor changes to the notation for the action sequence, replacing $a$ with $m$.

5. Corrected an inaccurate statement regarding SAC's effect on entropy for deeper actions in the action sequence.

6. Added benchmark results using the generative planning method (GPM).

7. Updated the FAS scores and associated plots for SRL. Previously reported values were based on stochastic actions during evaluation (specific to SRL). The revised values now reflect deterministic actions.

8. Presented separate plots for SRL and SAC for better clarity.

9. Added details on state-space and action-space dimensions to the online-planning results in Table 3.

---

> ### Author Response · Authors · 2024-11-28
> **Final updates to manuscript**
>
> To further address the concerns raised by reviewers YC71, Y33V, and pfCy, we have made the following updates to our manuscript:
>
> -Motivation Concerns: We have clarified and strengthened the motivation behind our work to address the points raised.
>
> -Results on TempoRL: We have added results for TempoRL to provide a more comprehensive comparison with existing methods.
>
> -Comparison Plots for SRL and SAC on the Same $J$: We included direct comparison plots of SRL and SAC on the same evaluation metric to enhance clarity and enable better assessment of their performance.
>
> -SAC with Randomized Frame-Skipping: We provided results for SAC with randomized frame-skipping across three environments, adding depth to the discussion of alternative approaches.
>
> We believe these updates have significantly improved the quality and rigor of our paper. We sincerely thank all reviewers for their constructive feedback and exemplary participation throughout this discussion phase.
>
> If there are any remaining concerns or further clarifications needed, we would be happy to address them.

---

### Author Response · Authors · 2024-11-30
**Thank You Reviewers!!**

We would like to express our heartfelt gratitude to all our reviewers for their active participation in this discussion and rebuttal process. We feel truly fortunate to have had such engaged reviewers—observing the silence in the discussions of other papers highlights just how exceptional this experience has been.

We are deeply appreciative of the reviewers’ thorough evaluation of our work and their dedication of significant time to what is often an unpaid and thankless task. We hope that our contribution to research has been worthy of your time, and we aspire to pay it forward by dedicating ourselves to this process in the future.

As we have seen negative feedback online regarding the ICLR review process, we believe this forum stands as a positive example of what the process can achieve. Finally, we wish to emphasize that our reviewers are a source of inspiration to the AI research community. These discussions are vital for providing external perspectives, improving research communication, ensuring quality control, and advancing the field—not just in terms of methodology, but also in philosophy and rigor.

Thank you for your invaluable contributions to science and the broader research community.

---

### Meta-Review · Area_Chair_Bfr7 · 2024-12-21

**Metareview:**

This paper addresses the implicit requirement of current RL usage in real-world environments where observations and action frequencies are all assumed to be synchronized with the decision-making algorithm’s frequency, which is difficult to achieve in real time at a high frequency. The paper proposes to mitigate it by learning an open-loop action sequence control with a low decision frequency, where the action sequence is learned with a background planner. The experiments show that the proposed method achieves good performance with a low decision frequency on tasks where RL algorithms typically operate at a high decision frequency. As such, I recommend accepting this paper.

The paper seems to have missed a highly related work: Karimi et al. (2022) “Variable-Decision Frequency Option Critic.” But the claims of this paper still hold, introducing a learned model to learn the action sequence, whereas in the other paper, the open-loop control seems to be not learned.

**Additional Comments On Reviewer Discussion:**

The reviewers are generally in favor of accepting the paper. The authors engaged with the reviewers extensively and addressed the main concerns.

---

### Decision · Program_Chairs · 2025-01-22

Accept (Poster)